# 60 Years of Studies into the Initiation of Chromosome Replication in Bacteria

**DOI:** 10.3390/biom15020203

**Published:** 2025-02-01

**Authors:** John Herrick, Vic Norris, Masamichi Kohiyama

**Affiliations:** 1Independent Researcher, 3 rue des Jeûneurs, 75002 Paris, France; jhenryherrick@yahoo.fr; 2Laboratory of Bacterial Communication and Anti-Infection Strategies, EA 4312, University of Rouen, 76000 Rouen, France; 3Université Paris Cité, CNRS, Institut Jacques Monod, 75013 Paris, France; masamichi.kohiyama@univ-paris-diderot.fr

**Keywords:** cell cycle, ribonucleotide reductase, DnaA, post-translational modification, initiator, licensing factor, condensate, DNA replication

## Abstract

The *Replicon Theory* has guided the way experiments into DNA replication have been designed and interpreted for 60 years. As part of the related, explanatory package guiding experiments, it is thought that the timing of the cell cycle depends in some way on a critical mass for initiation, *Mi*, as licensed by a variety of macromolecules and molecules reflecting the state of the cell. To help in the re-interpretation of this data, we focus mainly on the roles of DnaA, RNA polymerase, SeqA, and ribonucleotide reductase in the context of the “nucleotypic effect”.

## 1. Introduction

François Jacob and Sidney Brenner proposed the *Replicon Theory* on the basis of observations that had accumulated until about 1962. These observations included, firstly, the demonstration by Maaloe and Hanawalt that de novo protein synthesis is required for the initiation of DNA replication in *Escherichia coli* [1] and, secondly, the work on sexual conjugation in *E. coli*. The conjugation between *Hfr* and *F*- strains entails the transfer of the chromosome starting from a genetically determined, unique point on the chromosome of *Hfr*. Jacob and Brenner made the assumption that the normal replication of the chromosome is similar to that occurring during conjugation. They postulated that replication starts from a unique site after a specific interaction between a factor (the initiator) and the start site (the origin of replication). After an oral presentation of their hypothesis in 1962, they published a note entitled the “*Replicon Theory*” in 1963 [2]. Coincidentally, in the same year, two important results supporting their hypothesis were published: firstly, a picture of the chromosome of *E. coli* undergoing replication [3], and, secondly, evidence from germinating *B. subtilis* for the sequential duplication of the chromosome from a unique site [4].

In order to test the predictions of the *Replicon Theory*, two groups, one in Paris (at the Pasteur Institute) and the other in Cambridge (at the Laboratory of Molecular Biology), started to isolate mutants that failed to initiate DNA replication. Such mutations are lethal unless they are in the form of conditional mutations such as those that confer pH sensitivity or temperature sensitivity. Starting from an *E. coli* thymine-requiring mutant in which DNA synthesis was easily shown by its incorporation of radioactive thymidine, they isolated more than 100 mutants that grew at 30 °C but not at 42 °C. Around 10 mutants that failed to synthesize DNA at 42 °C were classified into those in which incorporation stopped immediately after the temperature shift to 42 °C and those in which incorporation only stopped after a delay.

Significantly, two of their mutants (*CRT-46* and -*83*) arrested thymidine incorporation at 42 °C with similar kinetics to that occurring at a permissive temperature in the presence of an inhibitor of protein synthesis, chloramphenicol (which echoed the earlier finding about initiation’s need for protein synthesis [1]). Moreover, these two mutants allowed lambda phage, after infection, to resume DNA synthesis at high temperatures, whereas the other eight mutants failed. This could be explained by the *Replicon Theory*, which proposes that each replicating unit possesses its own initiator. The fact that phage replication was independent of the replication of the host chromosome (in addition to differences in the kinetics of DNA synthesis) in *CRT-46* and *-83* supported the idea that these mutants were defective specifically in the initiation of chromosome replication [5]. Subsequently, these two mutants were mapped, and the mutated gene was called *dnaA* [6].

The *Replicon Theory* postulated that the nature of the initiator is independent of the enzymes responsible for DNA chain elongation. In the rolling circle model for *phi* X174 phage DNA replication, a nickase can be the initiator [7]. Accordingly, a *dnaA83* culture was briefly irradiated with gamma rays, which introduced nicks into the chromosome (and, less frequently, double-strand breaks). This irradiation provoked a new round of DNA synthesis, even at non-permissive temperatures [8]. This result shows that the action of DnaA as an initiator is independent of the system responsible for polymerizing DNA.

In parallel with the above, Helmstetter and his collaborators were laying the foundations for the modern understanding of the bacterial cell cycle based on the “membrane-elution” method to, for example, collect successive populations of new-born cells after previous labeling (or simply as a way to obtain a population of minimally perturbed cells at the same stage of the cell cycle) [9]. Cooper and Helmstetter found that the cell cycle of *E. coli B/r* growing rapidly (with doubling times from 20 to 60 min) could be described by two constants: C, the time to replicate the chromosome, and D, the time between the termination of replication of the chromosome and the completion of cell division (hence the C-period and D-period, respectively) [10]. In such fast-growing cells, rounds of replication overlapped so that there were no periods without DNA synthesis (unlike eukaryotic cells), while in slow-growing cells, there was a period—the B-period—without DNA synthesis between the birth of the cell and the subsequent replication of the chromosome.

The term “hyperstructure” has been coined to cover the wide variety of physical structures inside bacteria and organelles into which molecules and macromolecules are assembled [11,12], which is sometimes due to the phase separation that involves intrinsically disordered proteins and the intrinsically disordered regions of proteins [13,14,15]. In the hyperstructure approach, the initiation of chromosome replication depends on the assembly and functioning of a hyperstructure that comprises not just the basic orisome containing the classical replication enzymes but also the membrane, glycolytic, and other enzymes, as well as, via coupled transcription–translation, even the genes encoding the *Macromolecular Synthesis* operon [16]. The dynamics of this hyperstructure respond to a wide variety of inputs, including those from alarmones like (p)ppGpp, redox molecules like glutathione (GSH), and energy metabolites like ATP.

The term “nucleotypic effect” describes how, in eukaryotic cells, increasing the quantity of chromosomal DNA increases the size of the cells [17]. This has implications for cell cycle regulation in bacteria where increasing the quantity of chromosomal DNA (e.g., via increasing the growth rate by changing the media from poor to rich) also increases the size of the cells (which can more than double in mass) [18,19]. Here, we review many of the themes in cell cycle studies and discuss the problems they have raised, which we suggest may be addressed in the framework of the operation of an initiation hyperstructure.

## 2. In Vitro Studies

In principle, in vitro experiments allow a system to be stripped to its bare essentials. However, this can be at the expense of only having the downstream part of a pathway with the fundamental upstream part being absent. If some factor is essential to this upstream part, it may, therefore, be dismissed as non-essential. Moreover, a factor may be recruited to the initiation hyperstructure that is not essential for initiation but that plays an essential role in the subsequent replication. As this role may be a long way downstream of initiation, this factor may also be dismissed. Finally, a factor may be dismissed because it is only needed to counter the inhibitory action of another factor, despite the latter factor being important in vivo.

Most proteins cannot penetrate *E. coli*, which means that mutants defective in replication functions cannot be complemented in vivo by the addition of wild-type proteins. Hence, in order to isolate the DnaA protein or other replication factors, an in vitro DNA replication system is required. Obtaining a cellular fraction that reflected in vivo chromosome replication turned out to be difficult. For example, in vivo, DNA synthesis in the *CRT-266* mutant (later identified as a *dnaB* mutant) stops replication immediately after shift to 42 °C, while in vitro synthesis of a cellular fraction of this mutant was similar to that of the wild-type parent [20].

### 2.1. A Partially In Vitro System: Toluene-Treated Bacteria

One way of getting an in vitro replication system that reproduces in vivo conditions and that can receive additional macromolecules is to transform cells that lack the barrier of the cell wall. To obtain such a system, Bonhoeffer’s group invented the penicillin agar method [21]. Another way is to permeabilize *E. coli* by a solvent such as ether or toluene. *CRT-266* cells (*dnaBts polA*+) briefly treated by toluene can incorporate tritiated TTP in the presence of other three dNTPs and ATP at 30 °C but not at 39 °C; two intriguing results were that the addition of cell extracts from the wild type did not restore DNA synthesis at the higher temperature, and that this replication system required the high concentration of 1 mM of ATP [20].

### 2.2. The ColE1 Plasmid In Vitro Replication System

An in vitro DNA replication system was first made by Tomizawa’s group using cell-free extracts prepared from *E. coli* that replicate ColE1 plasmid DNA in vitro [22]. Fortunately, the replication of this plasmid depends on DNA polymerase I, and a simple lysate from cells containing ColE1 could, therefore, perform the reaction in the presence of the precursors of DNA. Since *E. coli* grows normally without DNA polymerase-I [23], which is encoded by *polA*, one strategy to obtain the genuine replicase was to use *polA* mutants.

### 2.3. oriC-Dependent In Vitro Replication Systems

A few years later, *oriC*, the replication origin of *E. coli,* was cloned [24,25], and Kornberg’s group succeeded in making a system for the replication of the *oriC* plasmid in vitro. Their system had two important characteristics: the presence of cytoplasmic proteins at a high concentration and the polymeric crowding agent, polyethylene glycol (which promotes phase separation). This system replicated *oriC* plasmid DNA in the presence of ATP, 4rXTPs, 4dXTPs, and an ATP-generating enzyme [26]. The system obtained from a *dnaA polA* mutant was found to be incapable of performing the reaction, which allowed them to purify a protein that restored DNA synthesis. This protein was DnaA. This complementation test allowed them to purify the products of other genes involved in DNA replication (*DnaBC*, *DnaG*, *DnaE*, *DnaN*, etc.). They were even able to reconstitute replication systems with these purified proteins, in one case with RNA polymerase, RNase H, and Topo I [27,28], and, in another case, without RNA polymerase [29]. In addition to these essential components, small nucleoid binding proteins (HU and IHF) improved the synthesis of DNA in these systems.

## 3. The DnaA Protein

### 3.1. Domain Traits

The *dnaA* gene codes for a monomeric protein of 52 kD comprise four domains [30,31] (Table 1). In addition to several deletion mutants, 16 temperature-sensitive mutants have been isolated and their mutated sites identified; 10 of these mutants have double mutations [32]. Among the nine mutants of the domain IIIa, all seven mutated at the ATP-binding site had a second mutation in other domains as if the first mutation was lethal and suppressed by the second.

Domain I is important for the interaction of DnaA with other proteins, such as DnaB [33] or Dps [34]. Domain II is thought to be a linker domain within which no mutation has been reported. That said, the deletion of Domain II suppresses the phenotype of the *seqA*, *obgE* double mutant but without affecting its growth [35]. Domain IIIa has the ATP-binding site, which plays an important role in its interaction with *oriC* (see below). Two proteins, DnaA5 and DnaA46, with mutations in Domain IIIa have been shown to be deficient in ATP-binding [36]. Moreover, this domain serves as an anchor for the opened DUE (duplex-unwinding-element) of *oriC* [37]. Domain IIIb, which can be deleted without a pronounced effect on growth, allows DnaA to bind to the membrane [38]. Two other interesting characteristics of Domain IIIb are its cAMP-binding site [39] and the existence of an ATPase-deficient mutation (*R344A*) [40]. Curiously, cold-sensitive *dnaA* mutants are all mapped in Domain IIIb [41]. One interpretation would be that this domain allows the protein to respond to the physical state of the membrane, which depends on temperature. Domain IV serves for sequence-specific binding to DNA [42].

**Table 1 biomolecules-15-00203-t001:** Domain traits of the DnaA protein.

Region ^†^	Properties
I1	Dia interaction [43], Dps interaction [34]
II78	Can be deleted or replaced by YFP [44], deletion suppresses *seqA obgE* [35]
IIIa136	Walker ATP-binding motif; ssDNA-binding sites V211 and R245 [37], AAA+ Arg finger motif important for *oriC* interaction [45], lysine178 can be acetylated by YfiQ, leading to inactivation [46]
IIIb295	Loss of membrane-binding due to deletion [38], cAMP-binding site [39], ATPase mutant [40], cold-sensitive mutants: R334H, R42H, E361H [41]
IV374	Sequence-specific DNA-binding [30,47]

^†^ Number of the first amino acid according to [32].

### 3.2. DnaA Interaction with oriC

As predicted by the *Replicon Theory*, DnaA binds specifically to a 9-mer sequence called the DnaA box (TTATXCACA), which is present at five different sites in *oriC* [48]. Kornberg’s group found that the addition of ATP, but not ADP, activates this interaction, and, most importantly, that double-stranded *oriC* DNA is partially opened during the interaction as expected for initiation of replication [49].

### 3.3. Detailed Analysis of oriC Opening by DnaA In Vitro

#### 3.3.1. Loop-Back Model

Katayama’s group tried to analyse the mechanism of *oriC* opening by DnaA-ATP, step-by-step, using *M13* plasmid DNA (7.7 kb) containing *oriC* [50,51]. The opening of *oriC* was detected by digestion of the substrate with P1 nuclease specific for a single-stranded DNA. *oriC* (246 bp) contains four DnaA boxes (R1 and R4 with high affinity, and R2 and R5M with medium affinity), as well as several low-affinity DnaA-binding sites (R2, I1, I2, C3, C2, I3, and C1). The lower affinity-binding boxes allow continuous binding of DnaA over the whole region. Other important elements participating in the opening include the DUE, which contains three AT-rich, 13-mers, and an IHF-binding site.

According to the results of Katayama’s group, the first step in initiation is the binding of DnaA-ATP to R1 and R5M, and then, after DnaA-ATP binding to low-affinity sites, IHF bends *oriC* by 180° in such a way that the DUE faces the DnaA-ATP oligomers. This hypothetical scheme, later named the “loop-back model” (Figure 1)**,** was re-examined by considering the peptide structures of both Domain III and IV [51,52]. DnaA-ATP bound to single-stranded DNA around R1 or R5M opens the DUE and forms a complex with the T-rich single strand of the DUE attached to V211 and R245 of the Domain IIIa of DnaA. Then, the DnaB helicase is loaded at the opened site in order to continue unwinding (by forming a pre-replication complex). It should be noted that IHF is not essential for initiation in vitro, since its absence only disturbs normal initiation in vivo (see below).

#### 3.3.2. *oriC* Opening Without ATP

The importance of ATP in the opening of *oriC* by DnaA, as described above, raises the question of how the *dnaA5* and *dnaA46* mutants, which are deficient in ATP binding, can open *oriC*. Landoulsi and Kohiyama found that the DnaA protein can perform its role without ATP using conditions that differed considerably from those used previously [53]. Firstly, the *oriC* substrate was in the form of minicircles, which were negatively supercoiled (and, at 641 bp, one-tenth the size of those used previously), and which were prepared by over-night incubation at 15 °C in a glass capillary with ligase and with BET. BET introduces negative supercoiling. Secondly, the substrate and DnaA were incubated at 30 °C for 30 min, conditions in which most of the minicircles are opened. This reaction is less efficient than P1 nuclease sensitization performed during 3 min of incubation at 38 °C with ATP and may be unable to open multiple copies of *oriC* simultaneously. This would fit with the asynchronous initiation phenotype of the *dnaA46* mutant [34], which is presumably due to the slow opening of *oriC* at the permissive temperature reducing the probability of having all the copies opening at the same time.

This result is consistent with the presence of an active site on the DnaA protein other than the AAA^+^ATP-binding site; two domains, IIIa with its single-strand DNA attaching site and IV with its possible DnaA-recognition capacity, are candidates for allowing *oriC* opening without ATP. These sites may be thermolabile in the DnaA5 and DnaA46 proteins. We speculate that *oriC* opening by DnaA without ATP depends, firstly, on the spontaneous breathing of DnaA boxes or partial opening of DUE by structural constraints as occurring in negatively supercoiled *oriC* minicircles, and, secondly, on a single-stranded region being enlarged by the insertion of DnaA. This speculation is supported by the observation of Jimenez–Sanchez’s group that an increase in the culture temperature by 10 °C induces an extra cycle of DNA replication in *E. coli* that depends on both DnaA and the DUE of *oriC* [54]. This could be explained by the denaturation of the DUE by heat, followed by the insertion of DnaA.

Another explanation for *oriC* opening by DnaA without ATP was provided by Kaguni’s group. They found that DnaA protein prepared from a *dnaA5* or *dnaA46* mutant with a defective ATP-binding site failed to replace the wild-type protein to allow *oriC* plasmid replication in Kaguni’s in vitro replication system with purified proteins. Seeking factors that could reactivate these thermosensitive proteins in cell-free extracts, they isolated two heat shock proteins, DnaK and GrpE; pretreatment of DnaA5 in the presence of ATP (which is necessary for heat shock proteins to function) restores the initiation activity at 30 °C but not at 40 °C. This result led them to speculate that these heat-shock proteins reactivated DnaA5 by inducing conformational changes that promote *oriC* opening, performed by some domains of DnaA [55] (see below).

#### 3.3.3. Single-Molecule Analysis of *oriC* Opening

Strick’s group performed this analysis, which involved attaching one end of a 2 kb fragment of *oriC* DNA to a magnetic bead and the other to a glass plate in the presence of different reactants; a magnetic field exerts a force of 0.1 pN on the bead and, hence, on the attached *oriC* DNA [56]. Rotation of the bead twists *oriC* and makes a corresponding change to the position of the bead, which is a measure of the twist. They found that the addition of DnaA results in about three positive twists of *oriC* in the presence of ATP, but not of ADP.

This conformational change of *oriC* can be attributed to DnaA attachment to *oriC,* since this change did not occur in a DNA fragment without *oriC*. However, they were unable to follow the kinetics of the interaction. Apparently, the formation of the complex occurs suddenly at any time during incubation at 34 °C, and even during prolonged incubation overnight. In contrast, the change did occur with an *oriC* lacking the DUE, which indicates that only the interaction of DnaA with the DnaA boxes is important. This is understandable due to the absence of IHF in the reaction mixture (even if the “loop-back” model is correct). It would be interesting to use their technique to follow the kinetics of the interaction between DnaA and *oriC* in the presence of all the elements implicated in initiation.

### 3.4. Mechanisms Regulating Initiation

#### 3.4.1. DnaA-ADP and the Negative Control of Initiation

The number of chromosomes doubles just once in a normal cell cycle. As explained above, DnaA is activated to bind *oriC* by ATP but not by ADP. This observation has been used to develop several models for the negative control of replication initiation. The simplest of these models is based on the ATPase activity of DnaA itself: the *R344A* mutation, which inactivates this reaction, makes the protein hyperactive [40] (see above). Before this discovery, Katayama’s group had discovered a system called RIDA (Replicative Inactivation of DnaA) in which the Hda protein and the beta-clamp (DnaN) transform DnaA-ATP to DnaA-ADP during replication [57]. Moreover, Domain IIIa, which has the ATP-binding site, is important for the reception of ADP from Hda-ADP. Finally, an *hda* mutant has problems both with growth and with the synchrony of initiation [58].

#### 3.4.2. *datA* Locus

The *dat* locus, a 1 kb-long region located at 94.6 min on the *E. coli* chromosome, contains five DnaA boxes and one IHF-binding site. Consequently, this locus can bind significant numbers of DnaA molecules [59]. Katayama’s group found later that *datA* and IHF binding to DnaA stimulate ATP hydrolysis, which may be considered as a possible negative control mechanism of initiation [60]. The *dat* deletion mutant has both an abnormal initiation mass and an abnormal synchrony of initiation that might be attributed to an excess of available DnaA molecules compared with the wild type [61]. Surprisingly, Morigen et al. found that the asynchrony revealed by flow cytometric analysis of a rifampicin-treated culture of the *dat* mutant is not due to abortive initiations occurring after overproduction of DnaA [62] but rather to rifampicin-resistant initiation occurring at a low concentration of rifampicin (but not at high concentrations) [61]. They also observed differences between *datA* and *hdaA* mutants in their capacity for maintaining *oriC* minichomosomes: a *datA* mutant harbours an increased number of minichromosomes, whereas an *hdaA* mutant has less, consistent with differences in their control over initiation.

#### 3.4.3. DARS1 and DARS2

Since DnaA-ADP is unable to open *oriC*, the factors that replace the ADP by ATP on DnaA may participate in initiation control. Katayama’s group found two loci, termed DnaA-Reactivating Sequences, on the *E. coli* genome that catalyze this replacement in vitro: DARS1 (a 103 bp sequence) and DARS2 (a 455 bp sequence) [51,63]. DARS1 and DARS2 contain three and six DnaA boxes, respectively. The presence of either of these sequences on a low copy number plasmid decreases the initiation mass [64]. Finally, a very recent publication demonstrates that the deletion of all four elements (Hda, DatA, DARS1, and DARS2) does not cause major problems to either replication or division, indicating either that these changes are of limited importance to the cell cycle or that *E. coli* is able to adapt to their absence [65]. That said, under fast growth conditions, this mutant did have more asynchronous initiations compared with the parental strain.

#### 3.4.4. DiaA

The DiaA protein was isolated by Katayama’s group during its study of a suppressor of the over-initiation phenotype of the *dnaA* mutant [43]. DiaA has an affinity for Domain I of DnaA and stimulates its activity via oligomerization [66].

#### 3.4.5. Membrane and Phospholipid

Replication of the chromosome is blocked by the addition of cyanide [67], while replication is stimulated by the addition of an uncoupler [68]. This stimulation has been attributed to the efflux of protons from the cell that still occurs in the presence of an uncoupler, leading to the strand separation of a replication complex located in, or close to, the membrane [68]. Kornberg’s group found that the *E. coli* membrane or phospholipid can rejuvenate DnaA-ADP in vitro [69,70]. In vivo, a mutant defective in the synthesis of acidic phospholipids did not initiate replication normally [71], while membrane domains form at an early stage of chromosome segregation in the *E. coli* cell cycle [72].

#### 3.4.6. SeqA

The control over initiation could be based on the quality or the quantity of DnaA itself (see below, Section 5.2 “The DnaA-ATP:DnaA-ADP Conundrum”). Another mechanism for this control could be based on the accessibility of *oriC* to DnaA. The involvement of the membrane in this mechanism was found by Schaechter’s group [73], which showed that *oriC* only bound to the membrane when *oriC* was hemimethylated at Dam sites (GA-N5methyl-TC), which are separated by 10, 21, or 32 bp (corresponding to one, two, or three helical turns of the DNA), and which are more frequent in *oriC* than elsewhere on the chromosome [74]. Landoulsi et al. then showed that this membrane preparation inhibits in vitro replication of *oriC* plasmids only when *oriC* is in the hemimethylated state [75]. Subsequently, Kleckner’s group and, independently, Austin’s group [76] isolated the protein responsible for this interaction, and the former group identified the gene, which they termed *seqA* [77]. They demonstrated that SeqA binds preferentially to hemimethylated Dam sites rather than to fully methylated or unmethylated Dam sites. SeqA dissociation from *oriC* is due to the complete Dam-methylation of the latter [78,79]. Another important SeqA function is related to the replication hyperstructure, where SeqA tracks behind the forks [80] and perhaps brings together genes associated with DNA replication and repair [81], which helps organize the cohesion of daughter chromosomes [82], again using its affinity for hemimethylated GATC sites.

The inhibition of the replication of hemimethylated *oriC* plasmids by the SeqA protein has been shown by Leonard’s group to be due to its occupation of the DnaA boxes containing Dam sites (R5M, I2, and I3), which prevents DnaA from binding to these boxes. Indeed, in *dnaC* cells, where the initiation of replication is suspended at an early stage, they found that DnaA bound to the high-affinity R1, R2, and R4 boxes but not to the lower-affinity R5M, I2, and I3 sites. The absence of DnaA binding to the R5M box and I2 and I3 sites may also be related to their location close to the IHF- and Fis-binding sites, so it is not surprising that mutants affected in these binding sites have asynchronous initiations [79].

In vivo, *oriC* sequestration via its Dam sites lasts for around 1/3 of the cell cycle [83], after which SeqA detaches from the region. Since this sequestration relies on the lower-affinity DnaA boxes containing Dam sites, DnaA-ATP-to-DnaA-ADP ratios might influence the importance of sequestration. Consequently, systems based on the quality–quantity control of the initiator (e.g., the ratio of DnaA-ATP:DnaA-ADP) or of the targets of the initiator (e.g., of the methylation state of DnaA boxes) might co-occur in vivo to different degrees.

However, an important question remains as to how sequestration ends. One simple explanation is a spontaneous SeqA detachment on the basis of the equilibrium constant: a SeqA-Dam M+/− complex may be dissociated due to the accumulation of free Dam M+/− sites produced by the progression of the replication fork. Another possibility is the disappearance of the hyperstructure involving SeqA and other hemimethylated DNA-binding proteins. Graumann’s group, for example, reported that in *B. subtilis*, which lacks Dam methylation, a 270 kD complex exists that is associated with the membrane and that involves the YabA protein, which negatively regulates DnaA [84].

#### 3.4.7. Cell Cycle-Dependent *oriC* Sequestration

Membrane-associated, hemimethylated *oriC* sequestration activity varies according to the cell cycle: The activity is highest at initiation and gradually decreases during replication elongation [85]. This observation suggests the existence of a SeqA-anchoring protein associated with the cell membrane whose activity fluctuates with the cell cycle. Rothfield’s group later identified a protein solubilized from the membrane that stimulates the hemimethylated binding activity of SeqA [86].

#### 3.4.8. AphA, a Periplasmic Protein with Hemimethylated DNA-Binding Activity

Kleckner’s group observed that in the *seqA* mutant, *oriC* sequestration is not completely abolished but reduced only by half compared to the wild type [77]. This observation suggests the existence of more than one hemimethylated *oriC*-binding protein. Using footprinting and gel-shift assays, Reshetnyak et al. identified an acidic phosphatase, AphA, that specifically binds hemimethylated *oriC*. Using DNase footprinting assays, a comparison between wild-type and *aphA* mutant extracts resulted in the identification of specific binding sites on hemimethylated *oriC* DNA [87]. However, the mutant *aphA* does not display an initiation asynchrony phenotype. This observation might be explained by the 6-fold increase in SeqA in the *aphA* mutant membrane preparation compared to that of the control [88].

## 4. Role of RNA Polymerase in the Initiation of Replication at *oriC*

### 4.1. Absence of an RNA Primer

The addition of rifampicin to an exponentially growing cell culture inhibits initiation. This is contrary to the result obtained from Kornberg’s system with crude cell-free extracts (FII) for *oriC* plasmid replication, which is not sensitive to rifampicin. Moreover, a reconstituted system with purified proteins but without RNA polymerase allows for replication [29]. This result is particularly interesting because it reveals a significant difference between in vivo and in vitro observations. A simple explanation would be that, in vivo, RNA polymerase synthesizes primer RNA for the initiation of replication. However, no such primer RNA has yet been detected (see below) [89]. Moreover, the deletion of the promoters for the two genes around *oriC* (*gidA* and *mioC*) does not affect initiation [90].

### 4.2. RNA Polymerase-DnaA Complex

Bagdasarian et al. found that the rifampicin-resistant mutation of the *rpoB* subunit of RNA polymerase suppresses the thermosensitivity of the *dnaA46* mutant [91], which led to the idea of a direct interaction between RNA polymerase and DnaA. Subsequent publications of Skarstad’s group support this idea: firstly, they found that in the *datA* mutant, which overproduces DnaA, the inhibition of initiation or replication by rifampicin requires a high concentration of the drug [61], as if a DnaA-RNA polymerase complex were to become partially refractory to the drug; secondly, they were able to isolate a DnaA-RNA polymerase complex from an *oriC* in vitro system containing the cross-linker, disuccinimidyl suberate [92]_._

### 4.3. Inhibition of Initiation by (p)ppGpp

Glaser’s group showed that initiation of replication is inhibited by the accumulation of (p)ppGpp [93]. (p)ppGpp binds to sites on RNA polymerase, which inhibits both transcription [94] and DNA replication. It is also notable that (p)ppGpp can bind to the primase, DnaG, to affect replication [95,96,97] (see Section 4.5). Accumulation of (p)ppGpp in an exponentially growing culture results in a similar pattern to that obtained with rifampicin in flow cytometry analyses. This inhibition of the initiation of replication might, therefore, be due to the blocking of RNA polymerase by (p)ppGpp in a manner similar to that of rifampicin.

To investigate the importance of transcriptional activation in the initiation of replication, Laub’s group inserted the promotor for T7 RNA polymerase (which is insensitive to (p)ppGpp) on either side of *oriC* and examined the effect of (p)ppGpp. Since they found no inhibition of initiation by (p)ppGpp when T7 RNA polymerase progresses and increases the negative supercoiling of *oriC*, they concluded that the loss of negative supercoiling around *oriC* due to the inhibition of RNA polymerase by (p)ppGpp constitutes the mechanism of initiation inhibition [98].

It should be noted that the importance of negative supercoiling in initiation was shown many years ago by Filutowicz, who examined the effect of DNA gyrase inhibitors on a *dnaA46* mutant [99]. The (p)ppGpp result of Laub’s group might also be explained by their finding that the concentration of DnaA was significantly reduced, probably due to reduced transcription because of (p)ppGpp accumulation: this would have led to the DnaA boxes, R5M, I2, and I3, being occupied not by DnaA but rather by SeqA, the absence of which relieved this (p)ppGpp inhibition. Against this, these authors have shown that overproducing DnaA ectopically did not relieve the inhibition of by (p)ppGpp [98].

### 4.4. RNA Polymerase-Dependent oriC In Vitro Replication System

As mentioned above, Kaguni’s group succeeded in reactivating the DnaA5 mutant protein by a pre-treatment with heat shock proteins in the case of a reconstituted replication system with purified proteins (see above). After reactivation, the replication assay mixture, in fact, required the presence of RNA polymerase (in addition to essential purified proteins). This requirement for RNA polymerase was also found in the case of another mutant protein, DnaA46, which also lacks the ATP-binding site, but there was no such requirement for RNA polymerase in the case of the wild-type DnaA. One explanation would be that RNA polymerase forms a functional complex with the mutant DnaA5 when the latter undergoes conformational changes caused by heat shock proteins. Another explanation would be that RNA polymerase induces the negative supercoils around *oriC* that are required by DnaA5 or DnaA46 for initiation. Unfortunately, no data are available as to whether the initiation of replication of *oriC* in vitro using DnaA5 is sensitive to rifampicin.

### 4.5. De Novo Synthesis of DnaA Needed for Initiation

Riber and Lobner–Olesen obtained results consistent with the initiation arrest provoked by either rifampicin or (p)ppGpp in vivo merely due to the lack of availability of DnaA [100]. In fact, they found that de novo synthesis of the DnaA protein that followed transcription by T7 RNA polymerase relieves blockage by either of the above inhibitors. One may infer that most of the DnaA proteins in the cell occupy DnaA boxes, membrane sites, or hyperstructures, such that a renewed supply of the proteins is necessary for initiation. This might also be true for the arrest of initiation due to treatment with chloramphenicol.

## 5. “The Initiation Mess” Revisited: Conundrums

In view of the numerous condundrums that the concept of the “initiation mass” has given rise to, it has been argued that it might be better dubbed the “initiation mess” [101].

### 5.1. The In Vitro/In Vivo RNA Polymerase Conundrum

The absolute dependence on RNA polymerase during in vivo cell growth is the salient feature distinguishing the in vitro from the in vivo replication initiation system. It has long been assumed that RNA polymerase is essential because the in vivo system might depend on RNA polymerase to prime initiation for DNA synthesis at *oriC*, in the manner that Okazaki fragments require RNA primer to initiate lagging strand DNA synthesis. Evidence for an active role of RNA polymerase in the initiation of DNA replication resides principally in the observation that inhibiting RNA synthesis with rifampicin blocks DNA synthesis [102]. As mentioned above, no such primer has been identified as being essential for replication initiation at *oriC* [103].

Transcription from genes flanking *oriC*—*mioC* and *gidA*—modulates replication initiation by introducing positive or negative DNA supercoiling. However, the products of these genes themselves are not essential, and the transcription of these regions is not required for initiation. In contrast, a mutation in *rpoB* suppresses the *dnaA46* temperature-sensitive mutation, while a mutation in *rpoC* results in a two-fold-to-three-fold increase in DNA concentration in a replicon-dependent manner [104]. The behaviour of these two mutants is consistent with a direct interaction between RNA polymerase and DnaA at *oriC*, an interaction that would be essential in vivo but dispensable in vitro. However, the increase in the chromosome number might be explained by an overall effect on cellular transcription rather than RNA polymerase acting specifically at or near *oriC* [105].

Progression through the *E. coli* cell cycle requires de novo protein synthesis and de novo DnaA synthesis for each division cycle. De novo protein synthesis might explain the need for RNA polymerase in vivo but not in vitro, where the requisite biochemical components are in abundant supply. The in vivo dependence on de novo protein synthesis would seem to imply that essential metabolites are exhausted during cell growth, certain proteins and enzymes need to be recycled for key structures to be re-assembled (and/or disassembled), or that the process itself of translation is required (see below).

Interestingly, the increase in the chromosomal copy number in the *rpoC* mutant corresponds to a five-fold increase in DnaA [104], yet hyper-initiation in the mutant does not result in stalled or collapsed forks, in contrast to DnaA-induced hyper-initiation in wild-type and other strains that over-initiate DNA replication [62]. This might suggest that the *rpoC* mutant allows an over-expression of other genes whose products are required for fork stabilization/restart and/or replication elongation. A possible candidate is Ribonucleotide Reductase (RNR), which supplies dNTPs for DNA synthesis and fork progression (see below). Significantly, the expression of the *nrdAB* genes encoding RNR, for example, is upregulated during DNA replication stress in all organisms examined so far, indicating its importance in DNA repair and replication fork restart [106].

Mutations in *hda*, *dnaA,* and *dnaN* genes also increase the concentration of RNR [107], presumably by suppressing DnaA inhibition of *nrdAB* gene expression [108]. Mutations in *dnaN* are known to result in elevated levels of *nrdAB* expression in a strictly Hda-dependent manner [109], consistent with the observation that low DnaA-ATP levels de-repress *nrdB* expression, while high levels repress transcription and consequently limit/decrease RNR and dNTP pool sizes [110,111]. Furthermore, transcription rates of *dnaA* and *nrdA* oscillate during the cell cycle: *nrdA* transcription levels are maximal when *dnaA* transcription levels are minimal and vice versa (180 degrees out of phase, see Extended Data Figure 6B in [112]), indicating a form of cell size-dependent, regulatory cross-talk between the two genes.

The *rpoC* mutant might, therefore, affect DnaA repression/de-repression of *nrdAB* transcription (see also [109]). Additionally, mutations in *rpoC* affect growth rates in a nutrient-dependent manner that resembles the “stringent response” in minimal media [113]. These findings reveal a clear effect of the rates of transcription and protein synthesis on DNA/cell mass homeostasis [114].

### 5.2. The DnaA-ATP:DnaA-ADP Conundrum

Very recently, a fluorescence-based study found that the intracellular distribution of DnaA is homogeneous in contrast to the distribution of HU (and, by extension, DNA), which is heterogeneous with respect to the halves of the cell corresponding to the future daughters. These variations in the DnaA/DNA ratio in the future daughter cells were proposed to cause variations in the time of initiation [115].

It has been repeatedly suggested that fluctuations in the ATP/ADP ratio are crucial for timing replication initiation. However, only 15 to 30% of the more than 1000 DnaA proteins present in the cell are in the ATP form [116]. ATP pools must double during growth (×2.4) to maintain a constant concentration over the cell cycle. ATP pool sizes peak at the time of cell division and cell birth, and then they decline to a steady state level [117]. The initiation factor DnaA is active only when bound to ATP. The ATP/ADP ratios, in the form of DnaA-ATP/DnaA-ADP ratios (for example), might, therefore, determine the observed growth rate invariant cell mass per origin at which initiation occurs, the so-called “initiation mass” (*Mi*) [118]. However, this proposal faces several caveats. Put simply, it is difficult to envisage how cell mass, per se, could determine when in the cell cycle to initiate DNA replication [119].

While ATP concentrations are also independent of the growth rate and vary only 12% between slow and fast growth (compared to a variation in *Mi* of about 20%) [117,120], ATP/ADP oscillations seem unlikely to explain *Mi* and replication initiation timing, at least in a global regulatory role as a “timer” or “sizer” mechanism since ATP pools, for example, are at millimolar (from 1.2 to 3.6 mmol [117]) or supersaturating concentrations throughout the cell cycle [121]. The cellular need for excess ATP has been a longstanding mystery (see below).

In line with previous studies [122], the concentration of DnaA has recently been shown to be constant over the cell cycle [115]. DnaA has a high affinity for both ATP and ADP in the micromolar range with KD of 0.03 and 0.1 [49], suggesting that DnaA is bound to ATP throughout most of the cell cycle: [DnaA] = constant; [DnaA-ATP] = constant; and [DnaA-ADP] = constant at all growth rates and at all phases of the cell cycle [123]. Additionally, the DnaA-ATP-to-DnaA-ADP transition is not essential for wild-type initiation control except at high growth rates [65]. DnaA-ATP/DnaA-ADP ratios are, therefore, dispensable in determining *Mi* and initiation timing. The idea that the concentration of DnaA-ATP regulates initiation and determines the initiation mass is incompatible with the fact that the concentration of DnaA (saturated by the high concentration of ATP) is higher than needed for replication initiation (200–500/*oriC* versus 5–20/*oriC*) [59,124,125]. Since DnaA, like ATP, is continuously present at supersaturating levels, its level can be doubled without affecting initiation [126].

Another complication arises from the fact that DnaA is competent for initiation in the absence of ATP binding, although cell cycle regulation of its initiation competence is compromised. Mutations that result in hyper-initiation or asynchronous initiation, such as *dnaAcos*, *dnaA46,* and *hda*, all cause defects in either binding or hydrolyzing ATP, though the mutants remain allosterically competent for initiation [127]. The *dnaAcos* hyper-initiation phenotype is highly toxic to the cell and results in fork collisions that produce lethal DNA double-strand breaks [128], a problem not encountered in in vitro systems. The Hda-directed post-initiation hydrolysis of ATP would, therefore, serve primarily to inactivate DnaA that is already bound to *oriC*.

Yet another complication arises from the need to sequester *oriC* following initiation. DnaA-ADP detaches from *oriC,* allowing the sequestration protein SeqA to bind the hemimethylated DNA and block DnaA-ATP from rebinding *oriC* post-initiation. SeqA has been proposed to be the main negative regulator of initiation, but its role as an inhibitor is unclear since DnaA-ATP is inactivated by Hda and DnaA-ADP is already inert for re-initiation before SeqA binds hemimethylated *oriC*. SeqA’s role as a negative regulator of initiation would, in that sense, seem superfluous, but apparently it is not, and the two mechanisms (SeqA and Hda) presumably act in concert.

Sequestration lasts during approximately one-third of the cell cycle [129], suggesting that DnaA-ATP “licenses” *oriC* in the mother cell [130], but replicons do not fire until after cell division and following de novo protein synthesis in either the mother or the daughter cells. Titration of DnaA by *oriC* induces *dnaA* gene expression, implying that *dnaA* is derepressed by *oriC-* and other DnaA-binding sites in order to maintain a constant DnaA concentration during cell growth. Auto-repression apparently serves to maintain a constant ratio between the amount of DnaA and *oriC* copy number throughout the cell cycle and at different growth rates [131].

Interestingly, protein synthesis is necessary for the disassembly of SeqA foci [132], which might explain why protein synthesis is required for replication initiation in vivo but not in vitro [79]. SeqA must disengage *oriC* in the mother cell to allow DnaA-ATP licensing and cell division followed by replication initiation in the daughter cells. At the same time, protein synthesis is required to recharge DnaA with ATP [116]. Whether protein synthesis is required pre- or post-initiation is unclear.

### 5.3. The Chaperone–Hyperstructure Conundrum

Another problem facing the DnaA-*Mi* hypothesis concerns heat-shock proteins. The chaperones GroELS, when overproduced, induce over-initiation in the absence of elevated amounts of DnaA46 in a *dnaA46* mutant, which would appear to contradict the notion that the ratio of the cellular amount of DnaA protein to *oriC* determines initiation frequency [133]. The authors suggest that the “DnaA protein may regulate initiation independently of simple changes in concentration”, implying some other factor is involved. This point raises serious questions about the notion that it is a constant amount of DnaA-ATP proteins (number of DnaA-ATP molecules bound to *oriC*) that triggers initiation once *Mi* has been attained. If, as these findings indicate, another factor is required to trigger initiation, then what might that factor be?

An obvious candidate for that auxiliary role would be ATP binding to DnaA in wild-type cells, followed by Hda-mediated, ATP hydrolysis: GroELS might compensate for the absence of ATP binding in *dnaA* mutants and also for the hydrolysis functions of DnaA (intrinsic) and Hda (extrinsic). Katayama and Nagata further propose that DnaA might adopt a partially unfolded state, enabling initiation of replication at *oriC*. Subsequently, “the protein may shift to a stable, fully folded, and inactive phase”, presumably when bound to ADP [133]. ATP hydrolysis in that scenario inactivates DnaA by promoting a fully folded DnaA structure. However, this remains to be demonstrated.

The DnaA protein harbours intrinsically disordered regions. Recently, it has been shown that deletions in the linker region play a significant role in ATP binding [134]. One plausible explanation for how GroELS suppresses *dnaA46* involves a GroELS-mediated stress response that bypasses the need for ATP to promote initiator self-assembly into an active hyperstructure [135]. ATP is a hydrotrope that, like chaperones, can impede the formation of aggregates of misfolded proteins and help dissolve them [136,137,138]. Hence, DnaA-ATP might promote the assembly of *oriC*-dependent initiation hyperstructures, whereas DnaA-ADP might promote the disassembly and inactivation of these hyperstructures.

### 5.4. The SeqA, Dam, SAM Conundrum: The Bacterial Cell Cycle as a Redox Oscillator

The above points support the argument that, while DnaA-ATP is active and DnaA-ADP is inactive, the ratio between them does not determine the *Mi*, and, therefore, the principal role of the transition between the two forms—in conjunction with SeqA—is to prevent free DnaA-ATP from re-binding *oriC* until DNA becomes fully methylated [78,79,139]. SeqA binding to hemimethylated *oriC*, in the latter scenario, would serve primarily to coordinate DNA segregation, DNA fork movement, and nucleoid restructuring by inhibiting TopoIV [78,140,141,142]. Inhibition of initiation, therefore, might be a simple consequence of sequestration, while benefiting the cell with the adaptive advantage of maintaining a critical distance between multiple replication forks on a single chromosome until sister chromosomes are fully segregated.

The above proposal is supported by the observation that overproduction of SeqA increases the period of sequestration at *oriC* and delays nucleoid segregation and cell division [78], apparently without impeding fork movement. Because both segregation and cell division are inhibited under these conditions, the persistence of SeqA at *oriC* argues in favour of SeqA playing a regulatory role in licensing *oriC* upstream from nucleoid segregation and cell division (Figure 2). DnaA production might, therefore, be needed for the post-sequestration licensing of *oriC,* in addition to (or instead of) the actual firing of initiation. Dam methylation of the *dnaA* gene, for example, is required for its post-sequestration expression, which is necessary to replenish DnaA, and, in *dam* mutants, DnaA levels decline to one-third the level in wild-type cells [139]. Whether or not the disassembly of SeqA foci serves as a signal to trigger segregation and division remains unclear.

Since sequestration lasts during only one-third of the cell cycle, an important question arises: What factor or event is responsible for disengaging SeqA from hemimethylated *oriC*, which depends on protein synthesis? The *dam* methyltransferase gradually replaces SeqA as it dissociates from hemimethylated DNA; but under physiological conditions, the methyltransferase does not actively displace SeqA [143], which indicates that some factor other than the *dam* methylase itself is involved.

S-adenosylmethionine (SAM) oscillates during the cell cycle and peaks at the S-phase in *C. crescentus* [144]. This might suggest that, in *E. coli,* SAM could be involved in mediating the detachment of SeqA from *oriC*. SAM, for example, allosterically enhances the affinity and binding of *dam* methyltransferase to DNA [145,146]. Once SeqA disengages from *oriC*, DnaA-ATP, would immediately re-bind to *oriC* post-sequestration. What, then, might prevent DNA synthesis from firing prior to cell division at a fully licensed *oriC*?

In addition to SAM, a number of energy and redox metabolites involved in replication initiation such as ATP, NAD(P)H, GSH, and dNTPs fluctuate in phase with the cell cycle [144,147,148], suggesting that the bacterial cell cycle is a redox oscillator that couples, or gears, major cell cycle events to changing rates of cell growth (Figure 2) [149,150]. NAD(P)H drives DNA synthesis in the S-phase of the eukaryotic cell cycle, providing the electrons needed to reduce RNR and activate dNTP synthesis [151], which occurs via the GSH and thioredoxin pathways [152]. The Trx pathway is induced in *Caulobacter crescentus*, for example, immediately before replication initiation and oscillates in phase with the bacterial cell cycle [153], presumably in response to reactive oxygen species generated in the respiratory chain or in the environment [154,155]. Significantly, oscillation of the intracellular redox state during the cell cycle of *C. crescentus* controls the activity of NstA, a protein that inhibits the decatenation activity of TopoIV [156].

SeqA has long been considered an inhibitor of premature reinitiation. However, like NtsA in *C. crescentus*, SeqA specifically inhibits TopoIV, which is required for segregation and, consequently, for cell division [80,157]. One possibility is that SeqA participates in the timing of DnaA-ATP licensing of *oriC* (post-sequestration or during the late C-period) for the initiations that will occur in the subsequent cell cycles of the daughter cells, or in fast-growing cells before cell division. The asynchrony phenotype in *seqA* mutants might, therefore, reflect aberrant licensing and the anomalous premature, stochastic activation of initiation before chromosome segregation and cell division have been completed.

In the case of RNR, it was found that inhibiting DNA replication increases the synthesis of *nrd* mRNA and that this increase depends on protein synthesis [158]. Moreover, the quantities of both *nrd* mRNA and RNR protein increase during the *E. coli* cell cycle to reach a maximum at the time of the initiation of DNA replication [159], as does GSH, indicating that an expansion in dNTP pool sizes occurs at or before the onset of DNA synthesis; dNTP pools then decline fivefold during the cell cycle from the peak at or before initiation [121], consistent with cell cycle pool size oscillations of GSH and dNTPs [144]. In *C. crescentus,* which relies essentially on Trx as a reducing agent, the need to precisely control Trx abundance is crucial to cell growth and underlies the importance of redox regulation of cell cycle progression, presumably via DNA synthesis [153]. In *E. coli*, mutation of the gene encoding the thioredoxine-like protein, YbbN, results in over-initiation but does not affect replication in vitro, which has been attributed to its acting as a chaperone for three enzymes in the DNA polymerase 3 holoenzyme, namely DnaN, HolE, and a subunit of HolB [160]. Alternatively, the mutation might affect RNR and perturb replication elongation in vivo, thus resulting in the observed over-initiation.

GSH is essential not only for initiation, but it also plays a role in coordinating cell division with cell growth [144], which might explain the positive correlation between cell age at replication initiation and cell age at constriction. As previously suggested: “Some event before or at chromosome initiation triggers the onset of cell division” [161]. A critical level of GSH might be one of these factors. Interestingly, SAM is also required for cell division [162], suggesting that these metabolites might serve as signals activated at the initiation of DNA replication that eventually trigger cell division or, indeed, that simultaneously trigger the process leading to cell division [16].

### 5.5. The Conundrum of the Initiation–Elongation Correlation

It has long been noted that there is a negative correlation between the cell cycle frequency of initiation and the rate of elongation (replication fork speed): hyper-initiation (more than one initiation event per chromosome per cell cycle) results in slower replication forks and, conversely, slower replication forks result in hyper-initiation (excess replication forks) [163]. This observation appears to be universal: a negative correlation between origin density (initiation frequency per kilobase) and fork rate has repeatedly been reported since 1975 in yeast and metazoan cells [164]. Experimentally perturbing fork progression (slowing forks) results in the activation of “dormant” origins and a reduction in average replicon size (for a proposed mechanism, see [165]).

Hence, there is a clear homeostatic relationship between the frequency of initiation during a single cell cycle and the rate of replication fork elongation. DnaA might license initiation of DNA synthesis from *oriC* [166,167], but it cannot actually cause this initiation until dNTP pool sizes are permissive for DNA synthesis. This proposal is supported by the observation that dNTP pool sizes are sub-saturating for replication fork rates at all growth rates—in contrast to DnaA and ATP, which are maintained at supersaturating concentrations throughout the cell cycle [168]. Deoxynucleotide pool sizes may, therefore, be limiting not only for replication fork progression but also for the initiation of DNA replication at *oriC*. However, this hypothesis remains to be verified.

It has been shown that over-expressing the cell cycle-regulated enzyme RNR not only expands dNTP pool sizes but also accelerates fork rates and shortens the C-period of *E. coli* independently of the growth rate. Conversely, either inhibiting or titrating RNR reduces pool sizes, slows fork rates, and lengthens the C-period [163,168,169]. The reduced fork rate when RNR is titrated in vivo results in a three-fold increase in chromosomal DNA and, concomitantly, a threefold increase in cell size (Figure 3)—a phenomenon well known in eukaryotes and referred to as the “nucleotypic effect” in which the C-value (defined as the total amount of DNA contained within a cell’s haploid chromosome set) determines cell size [17]. The nucleotypic effect inverses the assumed relationship between cell mass and cellular DNA content: DNA synthesis and, therefore, DNA content, drive cell mass accumulation during the cell cycle.

Thymine auxotrophic strains of *E. coli* can grow in a steady state despite the concentration of thymine in the medium being limiting [170]. In these conditions, the time taken to replicate the chromosome is lengthened [171], which is accompanied by morphological changes occurring to the chromosome interpreted as corresponding to changes in the number of “replication positions”. This number, termed “nucleoid complexity”, has been defined by the ratio of time taken to replicate the chromosome to the generation time (c/τ) [19], in contrast to “chromosome complexity” (defined as the *oriC/ter* ratio). It could be argued that the increase in nucleoid complexity during thymine deprivation reflects not only the slowing of replication but also an increase in the number of initiations. Indeed, inhibiting elongation under a variety of experimental conditions induces *nrdAB* expression and elevates levels of RNR [172].

Cell size depends on the growth rate and the rate of elongation during DNA synthesis (duration of the C-period). Experiments have shown in all organisms that fork rate depends on dNTP supply or RNR activity. High levels of RNR shorten the C-period, and low levels lengthen it [159,168]. However, the growth rate is largely unaffected and, hence, cells continue to grow and increase their masses when fork rate is experimentally impeded [163,171,173]. At the same time, initiation will continue, increasing nucleoid complexity or the number of replication origins and forks per chromosome. The accumulation of DNA results in larger-than-normal cells. This resembles the nucleotypic effect in eukaryotes: polyploidy and larger genome sizes cause cell size and cell cycle duration to increase. An example in prokaryotes of a nucleotypic effect is the *eel* phenotye [162]: cells have multiple fully replicated chromosomes, and cell size is correspondingly larger (up to 700× normal cell size). In mother cells with impeded forks, initiation (Init) occurs at the same *Mi* in the daughter cells but shifts to earlier in the cell cycle (time after cell division), thus reducing or eliminating the B-period in the daughter cells.

## 6. The Nucleotypic Effect, *Mi*, and Growth Rate “Invariance”

RNR, as the regulator of elongation, and DnaA, as the regulator of initiation, may form what has been termed a “homeostatic pair” (Figure 4): the two main replication factors cooperate in setting and coordinating initiation at a growth rate-dependent *Mi* (either variant or invariant; see: [118,174] and references therein); that is, a cell mass that is necessary and sufficient to guarantee a growth-rate invariant DNA/mass ratio for each division cycle. If the nucleotypic effect applies to bacteria, then it would not be surprising that the amount of DNA and the amount of cell mass accumulate in tandem under different (physiological) growth conditions in order to supply equivalent DNA/mass ratios to daughter cells—whether or not *Mi* is growth-rate invariant.

Accordingly, the gearing of major cell cycle events to the cell growth rate in the daughter cell is set by the rate of replication elongation, or duration of the C-period, in the mother cell: fork rate in the mother cell determines initiation timing in the daughter cell cycle. Three important points support that notion: (1) cell size scales with the duration of the C-period; (2) the duration of the C-period depends on the level of RNR activity [168], indicating that fork rate is the major cell cycle regulatory parameter in bacteria; and (3) the initiation hyperstructure “falls apart” or disassembles in the absence of dNTP synthesis [132,169]. Therefore, dNTP synthesis is necessary for the assembly of the hyperstructure and for replication initiation. Indeed, it has been suggested that the “control of replication initiation is not merely responsive to cell mass and nutritional status but may also reflect the status of ongoing replication forks” [176].

## 7. Discussion

The rate of fork movement relative to the growth rate of the cell (tau_CYC_/tau) determines the amount of DNA in the cell. This means that a cell with a slower rate of fork movement (causing a longer C-period) relative to its growth rate has higher concentrations of *oriC* and/or chromosomal DNA and, in consequence, is proportionally larger in size (the nucleotypic effect). This is consistent with a hypothesis in which the rate of fork movement in the mother cell influences the cell mass and the time at which initiation occurs in the daughter cell. In other words, the amount of DNA (complete, fully replicated, chromosomal equivalents) in the individual cell is a determinant of its mass and, hence, the amount of DNA is a determinant of the initiation mass, *Mi*. In this hypothesis, the *Mi* could, therefore, be considered more of a consequence or a correlate of initiation rather than its actual cause.

How might a nucleotypic effect operate in bacteria? Ribosome synthesis, which is limited by the metabolic rate [177], controls the rate and amount of cell mass accumulation (protein synthesis). RNR, likewise, controls the rate and amount of DNA synthesis. It seems plausible then that the production of these two factors (among others) is coordinated via RNA polymerase, such that an initiation mass emerges from their parallel expression: cell mass accumulating concomitantly with DNA synthesis. As the chromosome is being duplicated, the cell volume expands to accommodate the increasing amount of DNA until a critical DNA/mass ratio is reached before cell division. It remains to be shown if and how RNA polymerase activity might coordinate DNA synthesis with protein synthesis, but the in vivo requirement of RNA polymerase might reflect the need to couple DNA synthesis to cell mass accumulation in a manner establishing a growth rate invariant DNA/mass ratio.

One way to test this hypothesis would be by performing the following experiment: simultaneous, long-term (>12 h) overproduction of DnaA and RNR should result either in cell lysis (programmed cell death) due to perturbed DNA/mass homeostasis, or in a DNA/cell mass homeostasis in which cell sizes are directly proportional to complete fully replicated chromosomal copy numbers, perhaps resembling filamentous cells [178]. An *E. coli* mutant deficient for SAM synthesis, for example, is defective in cell division but maintains an otherwise normal cell cycle. The extremely elongated cells (*eel* phenotype) result in filaments that are 700 times longer than the usual cell length/size [162].

A second way to test the hypothesis would be to examine the level of RNR activity and dNTP pool sizes in the *rpoC* mutant, the expectation being that dNTP pool sizes will be elevated two-fold in order to account for the two-fold increase in chromosome copy number in these cells [104]. It might also be possible to isolate mutants other than *rpoC* that have a higher-than-normal chromosomal copy number. Babu et al., for example, have shown that the delta-*hda* strain overexpressing *nrdAB* has an elevated number of genomes compared to the wild type [179].

A number of questions remain to be addressed concerning the *hda*, *dnaA*, and *dnaN* initiation hyperstructure and its regulatory inter-relationship with *nrdAB* expression and the coordination of replication initiation with elongation, presumably in the form of a hyperstructure involving *oriC* and *nrdAB*. These questions include:(1)Is the *nrdAB* gene co-located with *oriC* at initiation when the hyperstructure forms?(2)Although some evidence exists that *nrdAB* levels are elevated in the *hda* mutant in certain genetic backgrounds (but not wild type), the *hda* gene itself is found in only a limited number of bacterial species [109]. How, then, is dNTP synthesis coordinated with replication initiation in Hda-deficient species?(3)The *dnaAcos* mutant over-initiates because it is refractory to regulatory mechanisms [180], and this raises the question of why *nrdAB* is not overexpressed in this mutant in proportion to the number of excess forks. Overproducing RNR rescues *dnaAcos* and *dna46* mutants from the lethal hyper-initiation phenotype at non-permissive temperatures, indicating that dNTP levels in ATP-refractory *dnaA(ts)* mutants like *dnaAcos* are not elevated and are similar to the wild type. This observation also applies to the *hda* mutant in which overproducing RNR suppresses the lethal hyper-initiation phenotype at non-permissive temperatures.(4)If the *dnaA* gene is auto-repressed [181], does this mean that the negative feedback regulation in *rpoC* is either relaxed or bypassed since its DnaA pools are elevated five-fold [104]? However, DnaA-ATP is not mutated in this genetic background, and, therefore, it is not clear if its regulation of *nrdAB* is similarly relaxed or bypassed (since DnaA-ATP represses *nrdAB* transcription). The high levels of DnaA-ATP in *rpoC*, for example, should impede fork movement if high (five-fold) levels of DnaA-ATP repress *nrdAB* transcription and, therefore, limit local dNTP pool sizes at the replication fork.(5)Does the main difference between in vitro and in vivo replication systems lie in the difference between the hyperstructures that operate in these systems? For example, the in vitro enzymic hyperstructure, which is based on diffusion, may contain only a subset of the constituents required in the full in vivo hyperstructure, which is based on channeling. The latter hyperstructure may comprise not only replication enzymes but also membrane and genes via coupled transcription-translation, which would mean that the proper functioning of an in vivo initiation hyperstructure requires RNA polymerase. Moreover, in the scenario of the phenotype being determined at the level of hyperstructures, transcription and translation have been proposed to be fundamental to the interactions between hyperstructures that result in initiation [182].

Finally, we suggest that an integrative interpretation of the above may require a radically different conceptual framework based on physical chemistry [16], one example of which would be a clock dependent on changes in the water potential, ψ [183]. How this clock might drive hyperstructure dynamics [184] is the subject of our current investigations.

## 8. Conclusions

The regulation of the bacterial cell cycle has long been studied in the framework of the *Replicon Theory* and the “initiation mass”. In a critical examination of these studies, we revisit *oriC*-based replication systems and focus on the DnaA protein and RNA polymerase and their interaction networks, which include RNR, chaperones, DatA, DARS1, DARS2, DiaA, SeqA, AphA, phospholipids, (p)ppGpp, cAMP, ATP, ADP, and dNTPs. In raising questions about the relationships between these networks and the coordination of the initiation of chromosome replication with elongation, we invoke the importance of hyperstructures containing both genes and proteins. We propose a homeostatic relationship between DnaA and RNR linked to a redox cycle that underpins the operation of a nucleotypic effect in bacteria, whereby the rate of replication fork movement in the mother cell determines the initiation mass in the daughter cell relative to cell cycle timing. To test this, we suggest several experiments that could be performed. Finally, we suggest that the in vivo requirement for RNA polymerase is related in part to its potential role in coordinating and coupling ribosome expression and protein synthesis to *nrdAB* expression and DNA synthesis in a growth rate-dependent manner, thus guaranteeing a constant DNA/cell mass ratio during cell growth and division. This prospect opens the way for potentially fruitful investigations into the mechanisms regulating and integrating cell size, DNA-replication timing, replication fork rate, cell mass, and cell cycle regulation in bacteria.

## Figures and Tables

**Figure 1 biomolecules-15-00203-f001:**
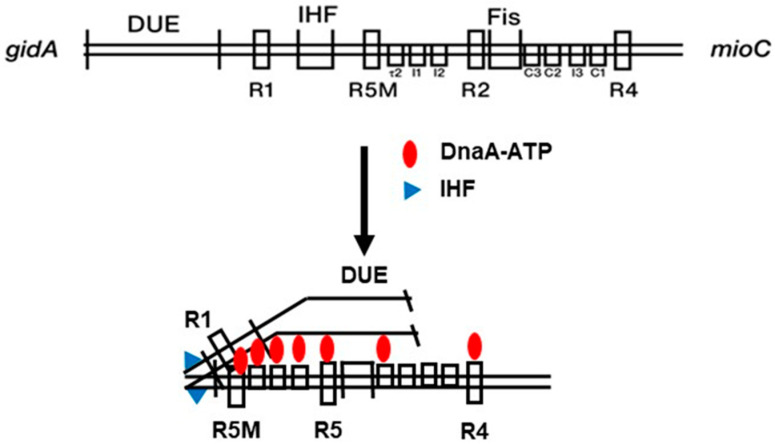
Loop-back model (adapted from [52]): *oriC* containing DUE (duplex unwinding element), DnaA boxes (R1, R5M, R2, and R4), IHF (and the IHF-binding site), and Fis (and the Fis-binding site) is folded by IHF facilitating interactions between DnaA and DUE.

**Figure 2 biomolecules-15-00203-f002:**
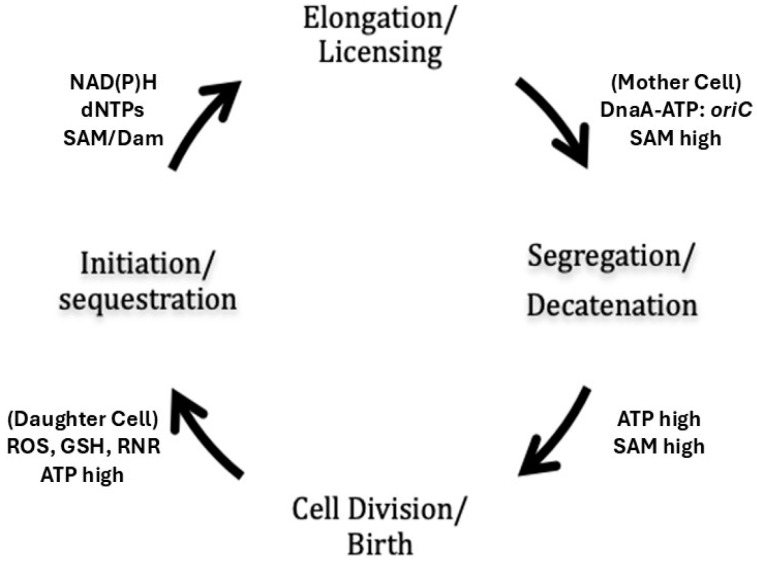
Mother cell licensing model. Licensing of *oriC* by DnaA-ATP occurs immediately after SeqA disengages from hemimethylated *oriC*, which is then rapidly methylated to form fully methylated DNA along the chromosome. During the C-period, *dnaA* expression is low due to elevated amounts of DnaA and auto-regulation of the *dnaA* gene. Elongation occurs at a constant rate depending on RNR activity and dNTP pool sizes at the replication forks. Topo IV is activated concomitantly with DnaA-ATP licensing of *oriC* after SeqA disengages from the fully methylated DNA (Elongation/Licensing). Segregation and decatenation then occur (Segregation/Decatenation). The mother cell divides, and fully licensed sister cells are born (Division/Birth). Protein synthesis, including DnaA synthesis and RNR synthesis, prime the licensed *oriC* for initiation when the right metabolic conditions are met (Initiation/Sequestration). The cycle is then repeated in the daughter cells. It is proposed here that licensing in the mother cell confers an adaptive advantage to the daughter cells by allowing them to respond rapidly to both external (temperature shift, etc.) and internal (stalled forks) challenges.

**Figure 3 biomolecules-15-00203-f003:**
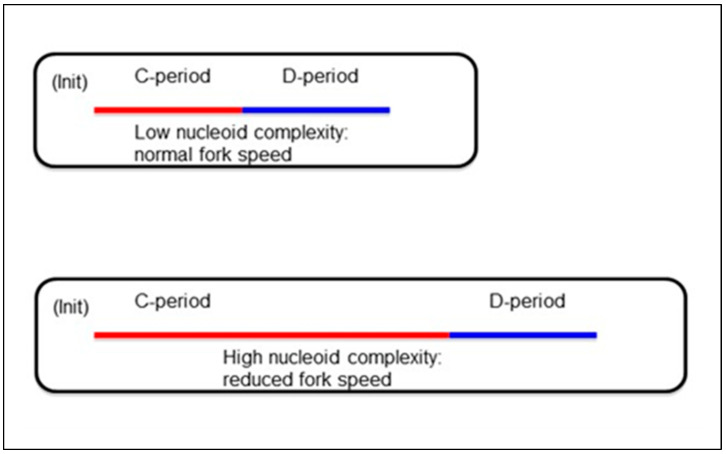
Model of the nucleotypic effect (adapted from [168]).

**Figure 4 biomolecules-15-00203-f004:**
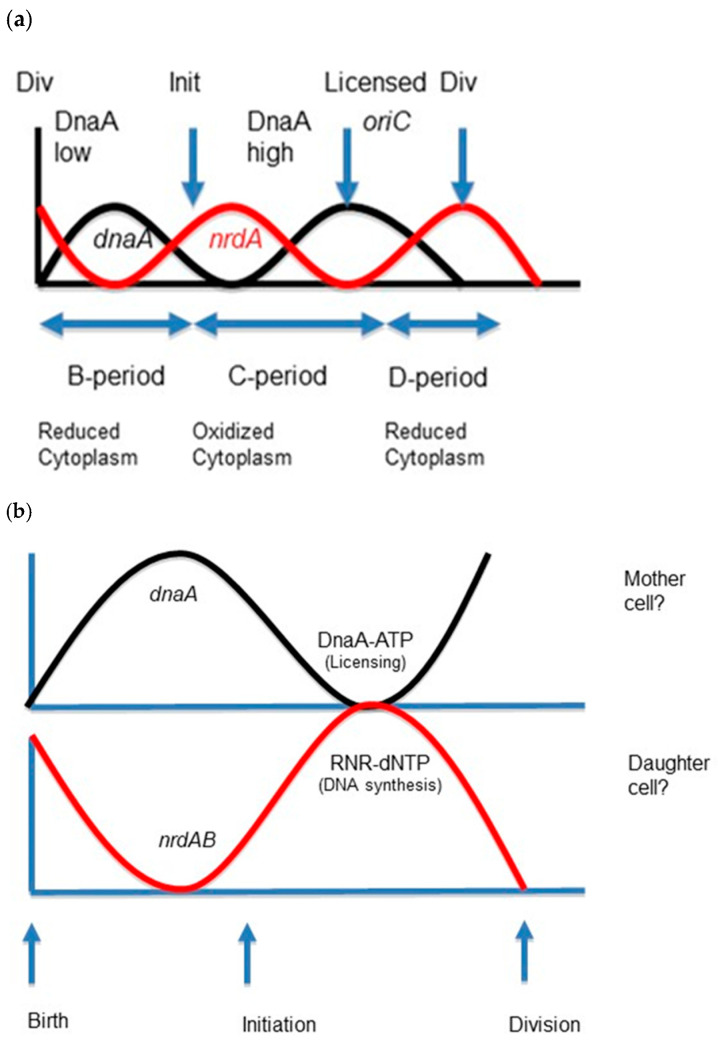
Model of DnaA:RNR homeostatic pair (adapted from [112]). (**a**) Expression of the *dnaA* gene (black) and the expression of the *nrdA* gene (red) oscillate out of phase in the first half of the cell cycle (see Extended Data Figure 6 in [112]): *dnaA* expression increases after cell division (div) when *nrdA* expression is low. Low levels of newly formed DnaA-ATP inversely correlate with high levels of *dnaA* expression (due to autoregulation) and stimulate *nrdA* expression. As DnaA levels increase, DnaA inhibits *nrdA* expression [110]. Consequently, DnaA protein will repress its own expression at or after initiation and will concomitantly limit RNR production by repressing *nrdA* expression during the S phase/C period, presumably to a level of expression that is in equilibrium with the number of active replication forks. DnaA and RNR thus act as a homeostatic pair to regulate the number of forks according to growth rate. The cytoplasm in *C. crescentus* is reduced in G1. At the G1/S transition (initiation of DNA replication), the cytoplasm switches to a highly oxidized state that becomes increasingly reduced during the S phase until G2 when the cytoplasm switches back to a reduced state [149,153,156]. This gated redox cycle also occurs in yeast and metazoan cells. It has been proposed to be a universal feature of cell cycle control [149,156] and, therefore, might explain the coordination between cell growth/mass and the major cell cycle events, including the initiation of DNA replication, which coincides with a relatively constant *Mi*. (**b**) Steps in the hypothetical relationship between licensing in the mother cell and initiation of replication in daughter cells. (1) DnaA binds *oriC* immediately after sequestration leading to its full licensing in late C- or D-periods; (2) licensing in the mother cell titrates DnaA prior to cell division; (3) low levels of DnaA at mother cell division and daughter cell birth induce *dnaA* expression in the B-period of the daughter cell cycle; (4) low levels of DnaA in the B-period of the daughter cell stimulate expression of *nrdAB* prior to initiation (late B- and early C-period); (5) high levels of DnaA in the early C-period repress *nrdAB* and limit RNR levels post-initiation (*nrdAB* expression is not necessary for elongation, but its expression is absolutely necessary for initiation, underlining the important role dNTP synthesis plays in the cell cycle regulation of replication initiation) [159,169]. An equilibrium (homeostasis) is established between rates of dNTP synthesis and the number of active forks, an equilibrium that serves to protect the cell against the mutagenic effects of abnormally high levels of RNR and/or rates of dNTP production (in conjunction with dATP allosteric regulation of RNR and NrdR repression of *nrdAB* [175]). It should be noted that during fast growth, there is no B-period. DnaA, therefore, licenses *oriC* in the mother cell (immediately after sequestration), and DnaA:RNR activates initiation prior to cell division. Black curve: *dnaA* gene expression level; red curve: *nrdAB* expression level.

## Data Availability

Not applicable.

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
