# Peer review of "60 Years of Studies into the Initiation of Chromosome Replication in Bacteria"

_biomolecules, 2025, doi:10.3390/biom15020203_

Round 1

Reviewer 1 Report

Comments and Suggestions for Authors

In the manuscript “60 years of studies into the initiation of chromosome replication in bacteria” by Herrick et al, the authors reviewed studies of the replication initiation using bacteria as the model system. The authors presented a comprehensive range of biochemical, genetic, and single-molecule studies on DnaA. They also explored the dynamic interactions between DnaA and the origins, or between DnaA and RNA polymerase. Notably, they described several challenges related to replication initiation, including the roles of RNA polymerase, the functions of DnaA when binding to ATP or ADP, and the involvement of chaperones. Overall, this manuscript offers a detailed summary of recent scientific advancements in bacterial replication initiation.

The authors are encouraged to incorporate the structural findings related to DnaA in their work. In particular, the recent cryo-electron microscopy (cryo-EM) structure of DnaA oligomers on double-stranded DNA (dsDNA) has provided valuable insights into how DnaA may interact with specific sequence elements to initiate DNA replication. Additionally, it's worth noting that gamma rays cause a significant number of DNA double-strand breaks, not just nicks, as mentioned on page 2, line 62 regarding gamma rays (which induce nicks in the chromosome).

Author Response

Comment1: In the manuscript “60 years of studies into the initiation of chromosome replication in bacteria” by Herrick et al, the authors reviewed studies of the replication initiation using bacteria as the model system. The authors presented a comprehensive range of biochemical, genetic, and single-molecule studies on DnaA. They also explored the dynamic interactions between DnaA and the origins, or between DnaA and RNA polymerase. Notably, they described several challenges related to replication initiation, including the roles of RNA polymerase, the functions of DnaA when binding to ATP or ADP, and the involvement of chaperones. Overall, this manuscript offers a detailed summary of recent scientific advancements in bacterial replication initiation.

Response1: We thank the reviewer for these positive comments.

Comment2: The authors are encouraged to incorporate the structural findings related to DnaA in their work. In particular, the recent cryo-electron microscopy (cryo-EM) structure of DnaA oligomers on double-stranded DNA (dsDNA) has provided valuable insights into how DnaA may interact with specific sequence elements to initiate DNA replication.

Response2: This sounds very important but we have been unable to find a paper reporting it.

Comment3: Additionally, it's worth noting that gamma rays cause a significant number of DNA double-strand breaks, not just nicks, as mentioned on page 2, line 62 regarding gamma rays (which induce nicks in the chromosome).

Response3: We have modified this sentence: “Accordingly, a dnaA83 culture was briefly irradiated with gamma rays, which introduce nicks into the chromosome (and, less frequently, double-strand breaks)”

Reviewer 2 Report

Comments and Suggestions for Authors

In this manuscript by Herrick and co-workers the authors present an overview of the investigations into how the initiation of DNA replication is regulated in bacteria. The authors have gone through a significant body of work and provide an overview of what emerges as an increasingly complex network of regulatory connections.

I very much enjoyed studying this manuscript. It reminded me of some of the older references, and it put both older and newer references in the context of our modern understanding. Having a comprehensive description summarising all this work is already an extremely valuable resource for a variety of researchers. The article is already very well developed, with very few grammar issues and typographical errors.

There are a few points raised below. However, these points are mostly aimed to increase quality of the work and accessibility by a larger audience.

Major point

Perhaps the weakest point of the manuscript is the question of what it wants to achieve. As a reference paper that comprehensively summarises 60 years of research into the topic, as the title suggests, I found it to be excellent. However, it does surprisingly little in improving the understanding of the issue at hand. Overall, it probably raises more question than it answers. There are some key points that I found to be very important, for example the notion that the cell size depends on the quantity of chromosomal DNA. But these get almost lost in all the descriptions of experimental results reported. At the very end when the reader is desperately looking for an answer, the authors hint at a possible solution that involves phase separation, but other than referring to a couple of recent papers that sound rather intriguing, nothing else is said, which left me as a reader rather disappointed.

One suggestion to improve the overall message of the work would be to very gently shorten the main body of the work. Too much shortening is probably not wise, otherwise the valuable role as a comprehensive overview of a lot of the literature might get lost. But some descriptions do not necessarily lead towards important new models, and here some shortening will be very helpful, as it will put emphasis on some of the results that are still relevant and perhaps under-explored. Shortening might also enable the authors to say a little more at the end about the new work that includes phase separation. It might make sense to exclude this, leaving the reader to refer to the original articles, but with phase separation being a relatively hot topic it would allow the authors to include another and potentially important angel into the discussion.

Minor points and corrections

Page 1, Line 24, I found the first sentence hard to read and understand. I would consider changing the phrasing to allow the opening statement to draw the reader in more easily.

P2, L70/72, E. coli needs to be italicised.

P2, L91/92, I found this entire discussion, which the authors also return to later, very important. I remember a number of discussions with various colleagues at conference debating the idea of cell mass as a trigger for initiation. One recurring argument is the significant different cell mass of slow and fast-growing bacteria. In minimal medium E. coli cells are about half the size of the cells routinely observed in rich medium, essentially excluding the notion that mass alone might be responsible. The notion of the quantity of chromosomal DNA being able to influence cell size explains exactly this discrepancy. I wonder whether inclusion of these strikingly different cells sizes in this context might be beneficial to other readers as well, as it puts the idea of the correlation of the quantity of chromosomal DNA and the cell size into context.

P3, L145, space before “H”, and which RNase are the authors referring to here? RNase HI or HII?

P6, L229, as the authors highlight that dnaA5 and dnaA46 are deficient in ATP binding at permissive temperature it might be helpful to the reader to quickly explain what is causing the temperature sensitivity of these two mutants.

P7, L279, delete the second colon at the end.

P9, when reading subchapters 4.2 and 4.3 I wondered briefly what exactly the physiological role for this interaction might and how it can be leveraged by the cell. It might be worthwhile highlighting here that an explanation will follow in the text below.

P12, L544-547, I had to read this sentence multiple times before I could decipher the meaning. Please clearify for the reader.

P19, L872, the text implies here that dnaA46 suffer from lethal hyper-initiation – the authors might want to clarify what is meant. dnaA46 cells grow perfectly fine at permissive temperature. Some clarification is needed here.

Author Response

Comment1: In this manuscript by Herrick and co-workers the authors present an overview of the investigations into how the initiation of DNA replication is regulated in bacteria. The authors have gone through a significant body of work and provide an overview of what emerges as an increasingly complex network of regulatory connections.

I very much enjoyed studying this manuscript. It reminded me of some of the older references, and it put both older and newer references in the context of our modern understanding. Having a comprehensive description summarising all this work is already an extremely valuable resource for a variety of researchers. The article is already very well developed, with very few grammar issues and typographical errors.

There are a few points raised below. However, these points are mostly aimed to increase quality of the work and accessibility by a larger audience.

Major point

Perhaps the weakest point of the manuscript is the question of what it wants to achieve. As a reference paper that comprehensively summarises 60 years of research into the topic, as the title suggests, I found it to be excellent. However, it does surprisingly little in improving the understanding of the issue at hand. Overall, it probably raises more question than it answers. There are some key points that I found to be very important, for example the notion that the cell size depends on the quantity of chromosomal DNA. But these get almost lost in all the descriptions of experimental results reported. At the very end when the reader is desperately looking for an answer, the authors hint at a possible solution that involves phase separation, but other than referring to a couple of recent papers that sound rather intriguing, nothing else is said, which left me as a reader rather disappointed.

One suggestion to improve the overall message of the work would be to very gently shorten the main body of the work. Too much shortening is probably not wise, otherwise the valuable role as a comprehensive overview of a lot of the literature might get lost. But some descriptions do not necessarily lead towards important new models, and here some shortening will be very helpful, as it will put emphasis on some of the results that are still relevant and perhaps under-explored. Shortening might also enable the authors to say a little more at the end about the new work that includes phase separation. It might make sense to exclude this, leaving the reader to refer to the original articles, but with phase separation being a relatively hot topic it would allow the authors to include another and potentially important angel into the discussion.

Response1: Our thanks to the reviewer for these – and many other – precious suggestions, which we have tried to take on board. We have shortened and modified section 4.2 RNA polymerase-DnaA complex: “Bagdasarian et al. found that the rifampicin-resistant mutation of the rpoB subunit of RNA polymerase suppresses the thermosensitivity of the dnaA46 mutant [91], which led to the idea of a direct interaction between RNA polymerase and DnaA. Subsequent publications of Skarstad’s group support this idea: firstly, they found that in the datA mutant, which overproduces DnaA, the inhibition of initiation or replication by rifampicin requires a high concentration of the drug [61] as if a DnaA-RNA polymerase complex were to become partially refractory to the drug; secondly, they were able to isolate a DnaA-RNA polymerase complex from an oriC in vitro system containing the cross-linker, disuccinimidyl suberate [92]..

We have also transferred section 3.4.9. De novo synthesis of DnaA needed for initiation to section 4.5.: “Riber and Lobner-Olesen obtained results consistent with the initiation arrest provoked by either by rifampicin or (p)ppGpp in vivo being merely due to the lack of availability of DnaA [100]. In fact, they found that de novo synthesis of the DnaA protein that followed transcription by T7 RNA polymerase relieves blockage by either of the above inhibitors. One may infer that most of the DnaA proteins in the cell occupy DnaA boxes, membrane sites or hyperstructures such that a renewed supply of the proteins is necessary for initiation. This might also be true for the arrest of initiation due to treatment with chloramphenicol.”

The reviewer makes a very important suggestion about adding new work that includes phase separation. Although we agree, the problem is that the more we add in terms of its explanatory potential, the more we need to add to justify it – and the more unbalanced the paper becomes. We have been through many versions trying to get something satisfactory and have failed. The solution is to write a follow-up paper focussing on phase separation and the water potential. To “improve the overall message of the paper”, we have added this paragraph:

“How might a nucleotypic effect operate in bacteria? Ribosome synthesis, which is limited by the metabolic rate [177], controls the rate and amount of cell mass accumulation (protein synthesis). RNR likewise controls the rate and amount of DNA synthesis. It seems plausible then that the production of these two factors (among others) is coordinated via RNA polymerase such that an initiation mass emerges from their parallel expression: cell mass accumulating concomitantly with DNA synthesis. As the chromosome is being duplicated, the cell volume expands to accommodate the increasing amount of DNA until a critical DNA/mass ratio is reached before cell division. It remains to be shown if and how RNA polymerase activity might coordinate DNA synthesis with protein synthesis, but the in vivo requirement of RNA polymerase might reflect the need to couple DNA synthesis to cell mass accumulation in a manner establishing a growth rate invariant DNA/mass ratio.”

Minor points and corrections

Comment2: Page 1, Line 24, I found the first sentence hard to read and understand. I would consider changing the phrasing to allow the opening statement to draw the reader in more easily.

Response2: We have changed the sentence to “These observations included, firstly, the demonstration by Maaloe and Hanawalt that de novo protein synthesis is required for the initiation of DNA replication in E. coli [1] and, secondly, the work on sexual conjugation in E. coli.”

Comment3: P2, L70/72, E. coli needs to be italicised.

Response3: Done

Comment4: P2, L91/92, I found this entire discussion, which the authors also return to later, very important. I remember a number of discussions with various colleagues at conference debating the idea of cell mass as a trigger for initiation. One recurring argument is the significant different cell mass of slow and fast-growing bacteria. In minimal medium E. coli cells are about half the size of the cells routinely observed in rich medium, essentially excluding the notion that mass alone might be responsible. The notion of the quantity of chromosomal DNA being able to influence cell size explains exactly this discrepancy. I wonder whether inclusion of these strikingly different cells sizes in this context might be beneficial to other readers as well, as it puts the idea of the correlation of the quantity of chromosomal DNA and the cell size into context.

Response4: Good idea! We have added this: “(e.g., via increasing the growth rate by changing the media from poor to rich) also increases the size of the cells (which can more than double in mass) [18,19])”

Comment5: P3, L145, space before “H”, and which RNase are the authors referring to here? RNase HI or HII?

Response5: We have corrected RNaseH to RNase H but RNase H I  or H II is not specified in the referred paper.

Comment6: P6, L229, as the authors highlight that dnaA5 and dnaA46 are deficient in ATP binding at permissive temperature it might be helpful to the reader to quickly explain what is causing the temperature sensitivity of these two mutants.

Response6: Good point. The dnaA46 and dnaA5 mutants, in which the protein lacks its ATP binding site, must use some other site(s) to open oriC. This site may be activated by the second mutation of DnaA46 or DnaA5 and may correspond to the conformational change caused by DnaK + GrpE addition found by Kaguni’s group. In our speculation, this oriC opening site would not operate at 40°C, as is the case for another temperature-sensitive enzyme, exonuclease III. We have added this sentence: “These sites may be thermolabile in the DnaA5 and DnaA46 proteins.”

Comment7: P7, L279, delete the second colon at the end.

Response7: Done

Comment8: P9, when reading subchapters 4.2 and 4.3 I wondered briefly what exactly the physiological role for this interaction might and how it can be leveraged by the cell. It might be worthwhile highlighting here that an explanation will follow in the text below.

Response8: We now tell the reader this in section 4.3. “(p)ppGpp binds to sites on RNA polymerase, which inhibits both transcription [94] and DNA replication; note too that (p)ppGpp can bind to the primase, DnaG, to affect replication [95-97] (see section 4.5.).”

Comment9: P12, L544-547, I had to read this sentence multiple times before I could decipher the meaning. Please clearify for the reader.

Response9: We have rewritten the sentence: “DnaA-ATP/DnaA-ADP ratios are therefore dispensable in determining Mi and initiation timing. The idea that the concentration of DnaA-ATP regulates initiation and determines the initiation mass is incompatible with the fact that the concentration of DnaA (saturated by the high concentration of ATP) is higher than needed for replication initiation”

Comment10: P19, L872, the text implies here that dnaA46 suffer from lethal hyper-initiation – the authors might want to clarify what is meant. dnaA46 cells grow perfectly fine at permissive temperature. Some clarification is needed here.

Response10: We have modified the sentence: “The dnaAcos hyper-initiation phenotype is highly toxic to the cell and results in fork collisions that produce lethal DNA double strand breaks [128]”

Reviewer 3 Report

Comments and Suggestions for Authors

This is a useful and timely update of the literature on the bacterial cell cycle, both old and current, — although the presentation could have been more orderly. But I learned a lot and had fun checking the literature, — although, as I said, the narrative tends to shift from one topic to another, so the reader has to keep the big picture in mind, not to be lost in the current section. Thus, the authors are urged to present the big picture early on and to return to it often.  

My comments are all minor, with the exception that the current understanding of the complexity of the ATPADPATP exchange in DnaA is missing. It needs to be adequately presented and incorporated into the overall initiation regulation scheme. This recent review should help (PMID: 29312202).

Comments

Page 5, top — yes, there are five major DnaA boxes at oriC, but the several minor ones (low-affinity I-sites, mentioned on line 206) allow continuous DnaA polymerization over this whole region (this was never mentioned).

Page 5, Fig. 1 — as is drawn, the scheme of oriC bending may be confusing, because both the left and the right sides of the top part change positions. I suggest to keep the right part (starting at the IHF binding site) unchanged, but bend (and open) the left side on it, like in the original scheme from Katayama’s paper.

         By the way, the reference in the legend is wrong — should be #50, rather than #49.  

Page 5, line 216 — “in order to resume DNA synthesis” should be deleted or replaced with “in order to continue unwinding”. No DNA synthesis is possible in the absence of the replisome (which is yet to be loaded).

Page 6, the entire section “3.3.2. oriC opening without ATP” — this whole section is unnecessarily speculative. The inability of the purified mutant DnaA proteins to bind ATP in vitro should not be equaled to the possibility that the mutant DnaA proteins somehow function without ATP in vivo. There will be other proteins in vivo that would compensate for the defect, making the mutant DnaA protein to take in ATP, — as the results with the DnaK and GrpE chaperones confirm. I suggest to remove this section completely as too speculative.  

Page 7, lines 295-296 — this information about the datA locus was later found to be incorrect. Katayama in 2013 clarified datA function as another place that promotes DnaA-ATP DnaA-ADP conversion — PMID: 23277577.

Page 8, the 3.4.6. SeqA section. The SeqA function at oriC could be described better (please consult PMID: 19254745):

— SeqA binding to unmethylated, full-methylated versus hemi-methylated GATC sites.

— SeqA binding to pairs of GATC sites, separated by 10,21,32 bp — and what it actually means in the context of “sister-chromatid chamber”.

— SeqA dissociation from oriC due to the complete Dam-methylation of the latter. (The authors seem to be aware of this — see page 13, lines 615-617)

— The eclipse period and its function. Lots of literature on this one.

Finally, it should be mentioned that another important SeqA function is at the “replication fork hyperstructure” (in Vic’s preferred term), where SeqA organizes the sister-chromatid-cohesion chamber behind the replication point, again using its affinity to hemimethylated GATC sites . The authors, again, mention this in passing later on (page 13, lines 617-619).   

Page 10, line 429 — paper #88 is about inhibition of transcription initiation — not replication initiation!

Page 10, lines 442-444 — yes, the decreased DnaA production is the other possible explanation for initiation block by pppGpp, — but the authors have addressed it by overexpressing DnaA or by showing that increased negative supercoiling at oriC can compensate even for the decreased DnaA.

Page 11, lines 510-513 — these speculations are not necessary and are mechanistically unsound. For example, rN incorporation is not known to affect the rate of DNA replication in bacteria (in vivo). Ref #102 describes purely in vitro experiments with purified replisomes, while Ref #103 describes the (very different!) situation in eukaryotes. 

Page 12, line 535 — 25 mM for ATP concentration must be an overestimate. The more realistic concentrations vary from 3 mM (your ref#111) to 10 mM.

Page 12, lines 538-541 — I am not sure how the first part of this sentence suggests the second part (that DnaA always binds ATP throughout the cell cycle). Besides, it is a direct contradiction of Katayama’s measurements, cited earlier (Ref. #106).

Page 12, line 555 — Ref #117 is a dud — while hyper-initiation does cause double-strand breaks at replication forks, — they are not caused by ROS. In general, it is preposterous (although fashionable these days) to claim endogenous oxidative stress in cells that are WT for ROS scavenging — this simply makes no sense for those who in vivo detect superoxide and hydrogen peroxide directly!  

Page 12, line 558 — DnaA is known to bind throughout the chromosome (>300 DnaA boxes) — so Had riding on the DNA clamp should be able to stimulate DnaA-ATP hydrolysis “globally”.

Page 14, lines 648-649: “mother cell confers an adaptive advantage to the daughter cells by allowing them to respond rapidly to a fluctuating environment either internally (stalled forks) or externally (temperature shift, etc.)” — This writing is awkward: it suggests that cells respond to the environment by either stalling their own forks (which, in principle, one could imagine) or by shifting external temperatures (???). Probably not what the authors meant…

Page 14, lines 651-656 — this paragraph is unnecessarily confusing. SeqA binding to hemi-methylated GATC sites is not absolute, and so Dam has a chance to fully methylate them, when SeqA dissociates. And once fully-methylated, GATC sites cannot be rebound by SeqA. It is not an active displacement, but a gradual elimination of SeqA binding sites.

Page 15, lines 720-721”until dNTP pool sizes are adequate to <…> complete DNA replication — the authors apparently assume that the cell must first accumulate enough dNTP to duplicate the entire chromosome, before initiating replication at the origin. At a minimum, they should 1) spell out this assumption clearly; 2) provide some evidence in support of it.

In fact, this is not true, because the limited size of the dNTP pools is known , and their usual size allows a very limited DNA synthesis indeed — as confirmed by the rapid inhibition phenotype of the dnaF(Ts) mutants at 42°C .

Page 15, lines 721-723 — the two statements about subsaturating dNTP pools versus supersaturating DnaA and ATP pools — need references.

Page 16, lines 742-744 — in addition to “nucleoid complexity”, there is also “chromosome replication complexity”, simply defined in bacteria as the ori/ter ratio .  

Miscellaneous

Page 6, line 274 — delete “of”

Page 9, line 426 — insert “transcription” before “initiation” — otherwise confusing

Page 10, line 460 — replace “replication” with “initiation”.

Page 11, line 507 — perhaps Fig. 4A? If both figures are referenced to the ref#101, then it should be Fig. 2h there.

Author Response

Comment1: This is a useful and timely update of the literature on the bacterial cell cycle, both old and current, — although the presentation could have been more orderly. But I learned a lot and had fun checking the literature, — although, as I said, the narrative tends to shift from one topic to another, so the reader has to keep the big picture in mind, not to be lost in the current section. Thus, the authors are urged to present the big picture early on and to return to it often. 

Response1: We thank the reviewer for the many helpful suggestions. Although we agree with the reviewer about the narrative shifting, our problem is that trying to give a unifying picture in terms of phase separation and the water-clock made the paper so complicated as to be unreadable. We have therefore limited our attempts to provide a big picture to expanding the ideas about the nucleotypic effect in the Discussion:

“How might a nucleotypic effect operate in bacteria? Ribosome synthesis, which is limited by the metabolic rate [177], controls the rate and amount of cell mass accumulation (protein synthesis). RNR likewise controls the rate and amount of DNA synthesis. It seems plausible then that the production of these two factors (among others) is coordinated via RNA polymerase such that an initiation mass emerges from their parallel expression: cell mass accumulating concomitantly with DNA synthesis. As the chromosome is being duplicated, the cell volume expands to accommodate the increasing amount of DNA until a critical DNA/mass ratio is reached before cell division. It remains to be shown if and how RNA polymerase activity might coordinate DNA synthesis with protein synthesis, but the in vivo requirement of RNA polymerase might reflect the need to couple DNA synthesis to cell mass accumulation in a manner establishing a growth rate invariant DNA/mass ratio.”

Comment2: My comments are all minor, with the exception that the current understanding of the complexity of the ATPàADPàATP exchange in DnaA is missing. It needs to be adequately presented and incorporated into the overall initiation regulation scheme. This recent review should help (PMID: 29312202).

Response2:  We now cite this helpful review in the sections where we discuss this cycling: 

3.3.1. “Katayama’s group tried to analyse the mechanism of oriC opening by DnaA-ATP, step-by-step, using M13 plasmid DNA (7.7kb) containing oriC [50] [51].” And again in section 3.3.1.: “This hypothetical scheme, later named the ‘loop-back model’ (Figure 1), was re-examined by taking into account the peptide structures of both Domain III and IV [51,52]."

3.4.3.: “Katayama’s group found two loci, termed DnaA-Reactivating Sequences, on the E. coli genome that catalyze this replacement in vitro: DARS1 (a 103 bp sequence) and DARS2 (a 455 bp sequence) [51,63].”

Please note that, as we say in two places in the text, “Finally, a very recent publication demonstrates that the deletion of all four elements (Hda, DatA, DARS1 and DARS2) does not cause major problems to either replication or division, indicating either that these changes are of limited importance to the cell cycle or that E. coli is able to adapt to their absence [65]; that said, under fast growth conditions, this mutant did have more asynchronous initiations compared with the parental strain. ” and “Additionally, the DnaA-ATP to DnaA-ADP transition is not essential for wild-type initiation control except at high growth rates [65].”

Comment3: Page 5, top — yes, there are five major DnaA boxes at oriC, but the several minor ones (low-affinity I-sites, mentioned on line 206) allow continuous DnaA polymerization over this whole region (this was never mentioned).

Response3: We have followed the reviewer’s advice:oriC (246bp) contains four DnaA boxes (R1 and R4 with high affinity, and R2 and R5M with medium affinity) as well as several low affinity DnaA binding sites (R2, I1, I2, C3, C2, I3, and C1); the lower affinity binding boxes allow continuous binding of DnaA over the whole region”

Comment4:  Page 5, Fig. 1 — as is drawn, the scheme of oriC bending may be confusing, because both the left and the right sides of the top part change positions. I suggest to keep the right part (starting at the IHF binding site) unchanged, but bend (and open) the left side on it, like in the original scheme from Katayama’s paper. By the way, the reference in the legend is wrong — should be #50, rather than #49. 

Response4: We have redrawn the figure and have corrected the reference, which we think should rather be to Wegrzyn and Konieczny (previously reference 52) where it was termed the “loop-back model”.

Comment5: Page 5, line 216 — “in order to resume DNA synthesis” should be deleted or replaced with “in order to continue unwinding”. No DNA synthesis is possible in the absence of the replisome (which is yet to be loaded).

Response5: Yes indeed! The sentence now reads “Then DnaB helicase is loaded at the opened site in order to continue unwinding (by forming a pre-replication complex).”

Comment6: Page 6, the entire section “3.3.2. oriC opening without ATP” — this whole section is unnecessarily speculative. The inability of the purified mutant DnaA proteins to bind ATP in vitro should not be equaled to the possibility that the mutant DnaA proteins somehow function without ATP in vivo. There will be other proteins in vivo that would compensate for the defect, making the mutant DnaA protein to take in ATP, — as the results with the DnaK and GrpE chaperones confirm. I suggest to remove this section completely as too speculative. 

Response6: It may be that the reviewer is assuming that DnaA5 or DnaA46 after DnaK and GrpE treatment becomes competent to use ATP for oriC opening. No published data are available for this assumption whereas opening oriC without ATP by DnaA in a particular setting has been reported. We would prefer to treat problems  in this review article that are based on the publications.

Comment7: Page 7, lines 295-296 — this information about the datA locus was later found to be incorrect. Katayama in 2013 clarified datA function as another place that promotes DnaA-ATP à DnaA-ADP conversion — PMID: 23277577.

Response7: We have taken the reviewer’s comment into account and have inserted the following phrase for the observation of Katayama’s group:

“Katayama’s group found later that datA and IHF binding to DnaA stimulate ATP hydrolysis, which may be considered as a possible negative control mechanism of initiation [60].”

Comment8: Page 8, the 3.4.6. SeqA section. The SeqA function at oriC could be described better (please consult PMID: 19254745):

— SeqA binding to unmethylated, full-methylated versus hemi-methylated GATC sites.

— SeqA binding to pairs of GATC sites, separated by 10,21,32 bp — and what it actually means in the context of “sister-chromatid chamber”.

— SeqA dissociation from oriC due to the complete Dam-methylation of the latter. (The authors seem to be aware of this — see page 13, lines 615-617)

— The eclipse period and its function. Lots of literature on this one.

Finally, it should be mentioned that another important SeqA function is at the “replication fork hyperstructure” (in Vic’s preferred term), where SeqA organizes the sister-chromatid-cohesion chamber behind the replication point, again using its affinity to hemimethylated GATC sites . The authors, again, mention this in passing later on (page 13, lines 617-619).  

Response8: We agree and have added to and amended this paragraph: “The control over initiation could be based on the quality or the quantity of DnaA itself (see below, Initiation Mass in 5.2.). Another mechanism for this control could be based on the accessibility of oriC to DnaA. The involvement of the membrane in this mechanism was found by Schaechter’s group [73] who showed that oriC only bound to membrane when oriC was hemi-methylated at Dam sites (GA-N5methyl-TC), which are separated by 10, 21, or 32 bp (corresponding to one, two or three helical turns of the DNA) and which are more frequent in oriC than elsewhere on the chromosome [74]. Landoulsi et al. then showed that this membrane preparation inhibits in vitro replication of oriC plasmids only when oriC is in the hemi-methylated state [75]. Subsequently, Kleckner’s group and, independently, Austin’s group [76] isolated the protein responsible for this interaction and the former group identified the gene, which they termed seqA [77]; they demonstrated that SeqA binds preferentially to hemi-methylated Dam sites rather than to fully methylated or unmethylated Dam sites. SeqA dissociation from oriC is due to the complete Dam-methylation of the latter [78] [79]. Another important SeqA function is related to the replication hyperstructure, where SeqA tracks behind the forks [80] and perhaps brings together genes associated with DNA replication and repair [81] helps organize the cohesion of daughter chromosomes [82], again using its affinity to hemi-methylated GATC sites.”

As regards the eclipse period, although we refer to it indirectly via reference 129, we would prefer not to discuss it here since it would risk taking us into lengthy speculations that some readers might find unpalatable.

 Comment9: Page 10, line 429 — paper #88 is about inhibition of transcription initiation — not replication initiation!

Response9: Yes, it is. Since we have already cited this paper (ref. 94), we have deleted this sentence (“In accord with this, rpoC or rpoZ mutants of RNA polymerase, which lack the (p)ppGpp binding sites, are unaffected by (p)ppGpp accumulation.”). We now say “(p)ppGpp binds to sites on RNA polymerase, which inhibits both transcription [94] and DNA replication; note too that (p)ppGpp can bind to the primase, DnaG, to affect replication [95-97] (see section 4.5.).”

Comment10: Page 10, lines 442-444 — yes, the decreased DnaA production is the other possible explanation for initiation block by pppGpp — but the authors have addressed it by overexpressing DnaA or by showing that increased negative supercoiling at oriC can compensate even for the decreased DnaA.

Response10: True. We have qualified our speculation by adding “Against this, these authors have shown that overproducing DnaA ectopically did not relieve the inhibition of by (p)ppGpp [98].”

Comment11: Page 11, lines 510-513 — these speculations are not necessary and are mechanistically unsound. For example, rN incorporation is not known to affect the rate of DNA replication in bacteria (in vivo). Ref #102 describes purely in vitro experiments with purified replisomes, while Ref #103 describes the (very different!) situation in eukaryotes. 

Response11: We have deleted the sentence to which the reviewer refers: “Alternatively, increased rNTP pool sizes resulting from slower transcription elongation might slow DNA replication fork progression via the incorporation of rNMPs, which is known to slow DNA replication elongation”

Comment12: Page 12, line 535 — 25 mM for ATP concentration must be an overestimate. The more realistic concentrations vary from 3 mM (your ref#111) to 10 mM.

Response12:  We have rewritten the sentence: “ATP pools, for example, are at millimolar (from 1.2 to 3.6 mmol [117]) or supersaturating concentrations throughout the cell cycle [121].”

Comment13: Page 12, lines 538-541 — I am not sure how the first part of this sentence suggests the second part (that DnaA always binds ATP throughout the cell cycle). Besides, it is a direct contradiction of Katayama’s measurements, cited earlier (Ref. #106).

Response13: We have modified the sentence to read: “DnaA has high affinity for both ATP and ADP in the micromolar range with KD of 0.03 and 0.1 [49], suggesting that DnaA is bound to ATP throughout most of the cell cycle: [DnaA] = constant; [DnaA-ATP] = constant; [DnaA-ADP] = constant at all growth rates and at all phases of the cell cycle [123].” 

Comment14: Page 12, line 555 — Ref #117 is a dud — while hyper-initiation does cause double-strand breaks at replication forks, — they are not caused by ROS. In general, it is preposterous (although fashionable these days) to claim endogenous oxidative stress in cells that are WT for ROS scavenging — this simply makes no sense for those who in vivo detect superoxide and hydrogen peroxide directly! 

Response14:  We agree and have rewritten the sentence, which now reads “The dnaAcos hyper-initiation phenotype is highly toxic to the cell and results in fork collisions that produce lethal DNA double strand breaks”

Comment15: Page 12, line 558 — DnaA is known to bind throughout the chromosome (>300 DnaA boxes) — so Had riding on the DNA clamp should be able to stimulate DnaA-ATP hydrolysis “globally”.

Response15: This comment perhaps refers to Hda removing DnaA-ATP as replication progresses along the length of the chromosome. To us, that would seem to suggest that Hda hydrolyses ATP “locally” at the fork rather than “globally” (“all at once”) upon initiation. This has been clarified by deleting “rather than globally hydrolyzing DnaA-ATP bound at other loci throughout the chromosome. This, however, remains an open question.”

Comment16: Page 14, lines 648-649: “mother cell confers an adaptive advantage to the daughter cells by allowing them to respond rapidly to a fluctuating environment either internally (stalled forks) or externally (temperature shift, etc.)” — This writing is awkward: it suggests that cells respond to the environment by either stalling their own forks (which, in principle, one could imagine) or by shifting external temperatures (???). Probably not what the authors meant…

Response16: We have rewritten the sentence: “It is proposed here that licensing in the mother cell confers an adaptive advantage to the daughter cells by allowing them to respond rapidly to both external (temperature shift, etc.) and internal (stalled forks) challenges.”

Comment17: Page 14, lines 651-656 — this paragraph is unnecessarily confusing. SeqA binding to hemi-methylated GATC sites is not absolute, and so Dam has a chance to fully methylate them, when SeqA dissociates. And once fully-methylated, GATC sites cannot be rebound by SeqA. It is not an active displacement, but a gradual elimination of SeqA binding sites.

Response17: Although we agree with this comment, how SeqA is removed from hemimethylated remains an open question. It is known that protein synthesis is required for SeqA removal. The rewritten sentence now reads:

“The dam methyltransferase gradually replaces SeqA as it dissociates from hemi-methylated DNA; under physiological conditions, however, the methyltransferase does not actively displace SeqA [143], which indicates some factor other than the dam methylase itself is involved.”

Comment18: Page 15, lines 720-721”until dNTP pool sizes are adequate to <…> complete DNA replication — the authors apparently assume that the cell must first accumulate enough dNTP to duplicate the entire chromosome, before initiating replication at the origin. At a minimum, they should 1) spell out this assumption clearly; 2) provide some evidence in support of it. In fact, this is not true, because the limited size of the dNTP pools is known , and their usual size allows a very limited DNA synthesis indeed — as confirmed by the rapid inhibition phenotype of the dnaF(Ts) mutants at 42°C .”

Response18: We do agree. We did intend to make it clear that dNTP pool sizes are sufficient to support only limited DNA synthesis (only a couple of minutes). It is evident however from the literature that the initiation hyperstructure involves RNR and dNTP synthesis but new or ongoing nrdAB/RNR synthesis is not needed to maintain the hyperstructure in an active state including at initiation (very beginning of DNA synthesis/C-period) See Guzman et al. 2002 Molecular Microbiology. The sentence now reads:

“it cannot actually cause this initiation until dNTP pool sizes are permissive for DNA synthesis. This proposal is supported by the observation that dNTP pool sizes are …”

Comment19: Page 15, lines 721-723 — the two statements about subsaturating dNTP pools versus supersaturating DnaA and ATP pools — need references.

Response19: We did cite references about this but, that said, we agree and have now added a reference:

 “This proposal is supported by the observation that dNTP pool sizes are sub-saturating for replication fork rates at all growth rates—in contrast to DnaA and ATP, which are maintained at supersaturating concentrations throughout the cell cycle [168].”

Comment20: Page 16, lines 742-744 — in addition to “nucleoid complexity”, there is also “chromosome replication complexity”, simply defined in bacteria as the ori/ter ratio . 

Response20:  We have modified the sentences: “This number, termed ‘nucleoid complexity’, has been defined by the ratio of time taken to replicate the chromosome to the generation time (c/τ) [19] in contrast to ‘chromosome complexity’ (defined as the oriC/ter ratio). It could be argued that the increase in nucleoid complexity during thymine deprivation”

Comment21: Page 6, line 274 — delete “of”

Response21: Done

Comment22: Page 9, line 426 — insert “transcription” before “initiation” — otherwise confusing

Response22: Yes, it is. We have rewritten the sentence and have added three references: “(p)ppGpp binds to sites on RNA polymerase, which inhibits both transcription [94] and DNA replication; note too that (p)ppGpp can bind to the primase, DnaG, to affect replication [95-97] (see section 4.5.).”

Comment23: Page 10, line 460 — replace “replication” with “initiation”.

Response23:  Done

Comment24: Page 11, line 507 — perhaps Fig. 4A? If both figures are referenced to the ref#101, then it should be Fig. 2h there.

Response24: We have corrected this to read: “(180 degrees out of phase, see Extended Data Figure 6B in [112])”

Reviewer 4 Report

Comments and Suggestions for Authors

This is a well-written and thoughtful review article on the mechanism of bacterial replication initiation. The authors provide a historical  perspective and discuss specific aspects of replication initiation and elongation in the context of a cellular "hyperstructure". The authors conclude with a discussion of the important issues and an outline of specific experimental strategies to address remaining questions.

Minor points:

1. A list of abbreviations would be helpful.

2. Instead of table 1, it may be better to show the protein structure of DnaA and highlight and describe the functional domains.

3. Line 462: Title "Initiation Mess", is this correct? The authors refer in this paragraph to "Initiation Mass".

Author Response

Comment1: This is a well-written and thoughtful review article on the mechanism of bacterial replication initiation. The authors provide a historical  perspective and discuss specific aspects of replication initiation and elongation in the context of a cellular "hyperstructure". The authors conclude with a discussion of the important issues and an outline of specific experimental strategies to address remaining questions.

Response1: We thank the reviewer for these encouraging remarks and for the suggestions.

Comment2: A list of abbreviations would be helpful.

Response2: Done

Comment3: Instead of table 1, it may be better to show the protein structure of DnaA and highlight and describe the functional domains.

Response3: The reviewer may be right but one of the other reviewers asked us to shorten the paper. We would therefore prefer to retain the Table and the new version of the figure showing DnaA’s sites and domains.

Comment4: Line 462: Title "Initiation Mess", is this correct? The authors refer in this paragraph to "Initiation Mass".

Response4: Yes, it is correct. We have now added the reference, Herrick et al. 1996, which is entitled “The initiation mess?”